# Statistical method scDEED for detecting dubious 2D single-cell embeddings and optimizing t-SNE and UMAP hyperparameters

Lucy Xia [1,7], Christy Lee[2,7] & Jingyi Jessica Li [2,3,4,5,6]

Two-dimensional (2D) embedding methods are crucial for single-cell data visualization. Popular methods such as t-distributed stochastic neighbor embedding (t-SNE) and uniform manifold approximation and projection (UMAP) are commonly used for visualizing cell clusters; however, it is well known that t-SNE and UMAP's 2D embeddings might not reliably inform the similarities among cell clusters. Motivated by this challenge, we present a statistical method, scDEED, for detecting dubious cell embeddings output by a 2D-embedding method. By calculating a reliability score for every cell embedding based on the similarity between the cell's 2D-embedding neighbors and pre-embedding neighbors, scDEED identifies the cell embeddings with low reliability scores as dubious and those with high reliability scores as trustworthy. Moreover, by minimizing the number of dubious cell embeddings, scDEED provides intuitive guidance for optimizing the hyperparameters of an embedding method. We show the effectiveness of scDEED on multiple datasets for detecting dubious cell embeddings and optimizing the hyperparameters of t-SNE and UMAP.

In the burgeoning field of single-cell biology, two-dimensional (2D) data visualization is an indispensable exploratory step that allows researchers to inspect the similarities and differences among single cells so as to discern the putative existence of discrete cell types and continuous cell trajectories. 2D data visualization is achieved by embedding methods, also known as dimension reduction methods. Among the many embedding methods developed for single-cell data[1–10], t-SNE[11,12] and UMAP[13,14] are the most popular because they can enhance the similarities of cells in the same type and increase the contrasts of disparate cell types.

However, the enhancement of similar cells' proximity in the 2D space comes at the cost of distorting the overall distances among all cells. In the trade-off between highlighting local cell clusters and preserving the global cell topology, t-SNE and UMAP focus on the local end, while the classic principal component analysis (PCA) sits at the global end by optimally preserving the global variance of cells. Despite the popularity of t-SNE and UMAP, cautionary messages have been raised against using t-SNE and UMAP embeddings to infer cell clusters' distances[15–19], which are related to the global cell topology.

[1]Department of ISOM, School of Business and Management, Hong Kong University of Science and Technology, Clear Water Bay, Hong Kong, China. [2]Department of Statistics and Data Science, University of California, Los Angeles, Los Angeles, CA, USA. [3]Department of Biostatistics, University of California, Los Angeles, Los Angeles, CA, USA. [4]Department of Computational Medicine, University of California, Los Angeles, Los Angeles, CA, USA. [5]Department of Human Genetics, University of California, Los Angeles, Los Angeles, CA, USA. [6]Radcliffe Institute of Advanced Study, Harvard University, Cambridge, MA, USA. [7]These authors contributed equally: Lucy Xia, Christy Lee. ✉e-mail: jli@stat.ucla.edu

Aligned with those cautionary messages, strategies have been proposed to optimize the hyperparameters of t-SNE and UMAP from the perspective of preserving the global cell topology[17,20–26], and new embedding methods have been developed to improve upon t-SNE and UMAP[8,15,27]. Nevertheless, none of these optimization strategies or new methods can guarantee to what extent the resultant cell embeddings will accurately preserve the distances between cell clusters. Hence, researchers are still perplexed about whether a particular cell cluster's neighboring clusters are trustworthy in the 2D-embedding space. A related question is whether distant clusters in the 2D-embedding space are truly distinct. Moreover, it remains questionable whether a visual cluster in the 2D-embedding space is trustworthy or should be divided into more than one cluster.

To address these questions, we present scDEED (single-cell dubious embedding detector), a statistical method that decides whether each cell's 2D embedding is dubious or trustworthy. The core idea of scDEED is to assess if a cell has similarly ordered neighbors up to a mid-range neighborhood before and after the 2D embedding (Fig. 1I). Based on this idea, scDEED assigns every cell a "reliability score," whose large value indicates that the cell's immediate to mid-range neighbors are well preserved after the embedding. Then scDEED compares each cell's reliability score to a null distribution of reliability scores (when all cells have random neighbors) and identifies the cell's embedding as dubious (or trustworthy) if the cell's reliability score is worse (or better) than 95% of the null reliability scores. The results from scDEED will help researchers avoid using dubious cell embeddings to interpret cell similarities and be confident about using trustworthy cell embeddings.

This dubious embedding detection functionality also gives scDEED another use: optimizing the hyperparameters of a 2D-embedding method by minimizing the number of dubious embeddings (Fig. 1II). Unlike existing optimization strategies for t-SNE and UMAP[17,20,21,23–26,28], scDEED offers users the flexibility to optimize hyperparameters in an intuitive and graphical way (users can see which cell embeddings are dubious under each hyperparameter setting), without modifying the embedding method's algorithm. Furthermore, scDEED's definition of dubious cell embeddings distinguishes scDEED from DynamicViz[26], a method that optimizes hyperparameters by minimizing the variance of cell embeddings' Euclidean distances across multiple bootstraps. Instead of checking the stability of 2D embeddings as DynamicViz does, scDEED evaluates the preservation of cells' immediate to mid-range neighbors in the 2D-embedding space.

Notably, scDEED is compatible with all 2D-embedding methods, and here we show its use for t-SNE and UMAP for demonstration purposes. This general applicability distinguishes scDEED from EMBEDR[25], a method that assigns every cell a quality score based on the t-SNE loss function and is thus unsuitable for comparing embeddings from different methods. In contrast, scDEED can compare embeddings from different methods at various resolutions, including individual cells, cell types, and all cells.

In this study, we applied scDEED to multiple scRNA-seq datasets. Our results show that scDEED successfully identified dubious cell embeddings in the original studies. Moreover, the hyperparameters optimized by scDEED resulted in cell embeddings that better preserved the biological relationships of cell types compared to the original studies. We also demonstrated the advantages of scDEED over EMBEDR in optimizing the perplexity hyperparameter of t-SNE, even though EMBEDR was designed based on the t-SNE loss function.

## Results
### A brief description of the scDEED method
Figure 1I illustrates the scDEED method, which evaluates a 2D-embedding method (such as t-SNE or UMAP) by comparing each cell's neighbors in the pre-embedding space and the 2D-embedding space, with both sets of neighbors defined by the Euclidean distance and a pre-specified neighborhood size, i.e., the number of cells multiplied by the "similarity percent" (scDEED's only hyperparameter, set to 50% by default; see Methods for an investigation of the similarity percent value). In practical uses of t-SNE and UMAP (such as in the Seurat pipeline), the pre-embedding space is defined by the top few principal components (PCs), whose number is usually between 20 and 50, of log-transformed normalized gene expression levels. As each cell's Euclidean distances to 2D-embedding neighbors are crucial for visual inspection of the data, scDEED defines for each cell a "reliability score" to measure the consistency of the cell's 2D Euclidean distances to ordered neighbors before and after the embedding. Specifically, two ordered distance vectors are constructed for each cell: a pre-embedding distance vector and a 2D-embedding distance vector, whose lengths are both equal to the neighborhood size. The pre-embedding distance vector contains the 2D Euclidean distances between the cell and the pre-embedding neighbors, following the order of these neighbors from the closest to the farthest in the pre-embedding space. The 2D-embedding distance vector contains the 2D Euclidean distances between the cell and the 2D-embedding neighbors, following the order of these neighbors from the closest to the farthest in the 2D-embedding space. Then, the cell's reliability score is defined as the Pearson correlation between the two vectors.

To construct a null distribution of reliability scores, scDEED employs a permutation strategy: independently permuting every gene's expression levels across cells. After the permutation, all cells' relationships are disrupted; all cells become exchangeable, and each cells' neighbors become a random set of all cells. As a result, all cell's reliability scores computed on the permuted data form a null distribution, i.e., the distribution of a cell's reliability score if 2D embedding disrupts the cell's neighbors in the pre-embedding space and randomly assigns neighbors to the cell in the 2D-embedding space (Methods). Based on this null distribution, two reliability score cutoffs are defined: (1) a trustworthy cutoff corresponding to the 95th percentile of the null distribution and (2) a dubious cutoff corresponding to the 5th percentile of the null distribution. Hence, for each cell in the original data, scDEED labels its 2D embedding as trustworthy if its reliability score is greater than or equal to the trustworthy cutoff. Conversely, if the cell's reliability score is less than or equal to the dubious cutoff, its 2D embedding is labeled as dubious. Cells that do not meet these criteria remain unlabeled. It is worth noting that the percentage of cell embeddings labeled as trustworthy (or dubious) is not necessarily 5%, as the percentile cutoffs are defined based on the null distribution rather than the reliability scores computed from the original data.

scDEED's identification of cell embeddings as dubious or trustworthy allows users to identify potentially spurious cell clusters in the 2D-embedding space that are artifacts of the embedding process rather than representing biologically meaningful cell types. It can also highlight cell clusters whose global positioning might be misleading. The lack of global preservation and the random positioning of clusters is a well-known issue in t-SNE and UMAP[20]. By identifying dubious cell embeddings, scDEED aims to address this issue. In the following sections, we will show examples of how the identification of dubious cell embeddings helps reveal dubious cell-type relationships in the 2D visualization.

To help users obtain a more trustworthy 2D visualization of data, scDEED provides an approach to optimize a 2D-embedding method's hyperparameters (e.g., perplexity for t-SNE; min.dist and n.neighbors for UMAP) via a grid search for the hyperparameter setting that minimizes the number of dubious cell embeddings.

We note that scDEED is a flexible method applicable to any 2D-embedding method, not limited to t-SNE and UMAP. In our results, we focus on t-SNE and UMAP to showcase the effectiveness of scDEED in enhancing the reliability of 2D visualizations for drawing biologically meaningful conclusions.

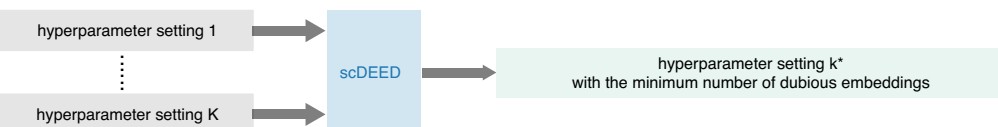

**Fig. 1 | Illustration of the two functionalities of scDEED.** Functionality I decides whether each cell has a *trustworthy* or *dubious* embedding by calculating a *reliability score*, which is defined as the Pearson correlation between the cell's distances to its closest 50% neighboring cells in the 2D-embedding space and the same cell's distances to its closest 50% neighboring cells in the pre-embedding space (with the distances in each space ordered from the 1st neighbor to the $[n/2]$th neighbor, where $n$ is the total number of cells). Compared with a *null distribution* of reliability scores, obtained through permutation, cell 1's reliability score (marked by the purple star) falls into the highest 5%, so it has a trustworthy embedding; in contrast, cell 2's reliability score (marked by the orange star) falls into the lowest 5%, so it has a dubious embedding. Enabled by functionality I, functionality II optimizes the hyperparameter setting of an embedding method (e.g., t-SNE or UMAP) by minimizing the number of dubious embeddings.

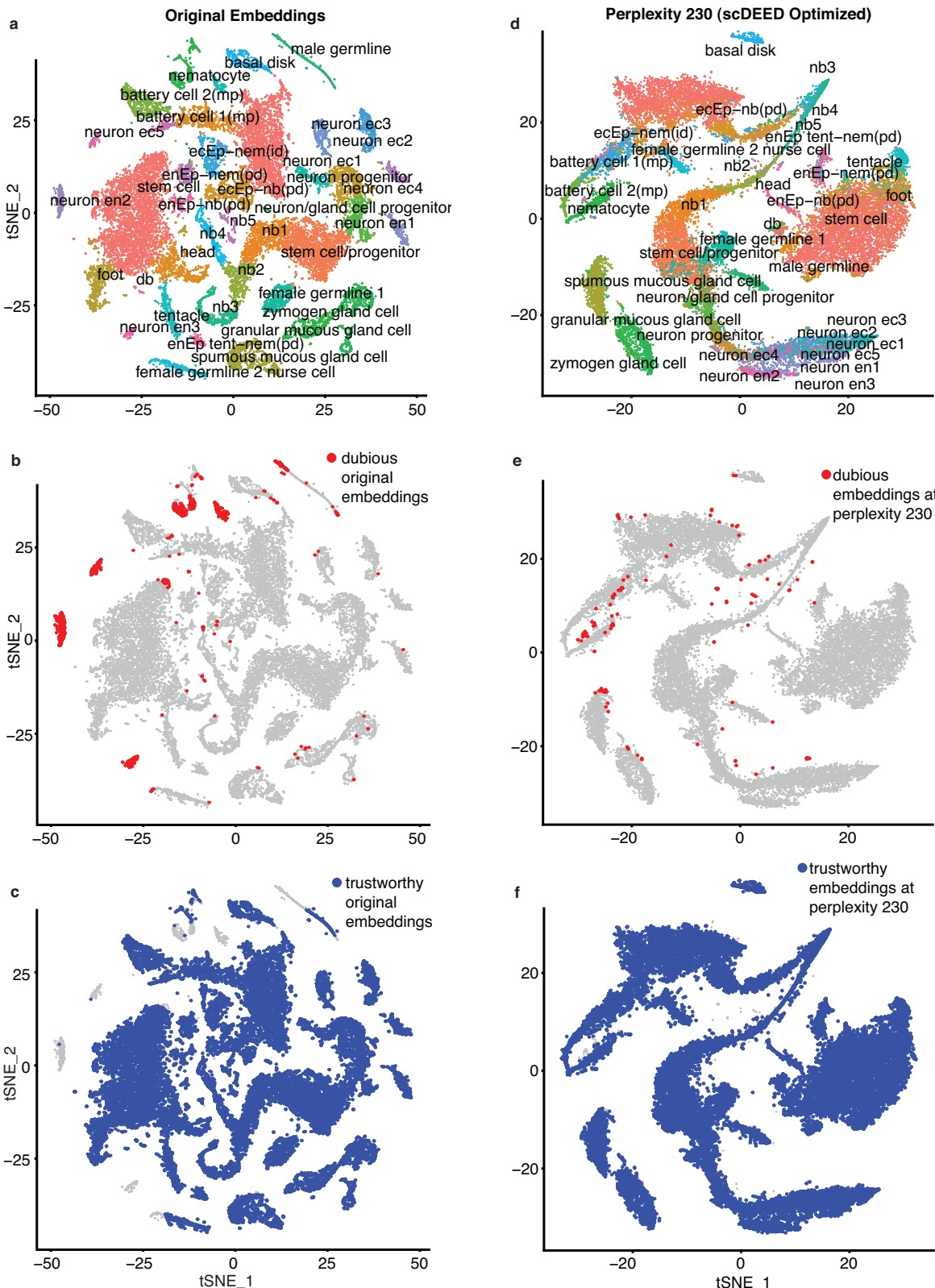

**Fig. 2 | Original t-SNE embeddings and t-SNE embeddings optimized by scDEED on the Hydra dataset. a** t-SNE plot of the Hydra dataset at the perplexity 40 used in the original study. **b–c** Dubious cell embeddings (**b**) and trustworthy cell embeddings (**c**) defined by scDEED at the perplexity 40. **d** t-SNE plot of the Hydra dataset at the perplexity 230 optimized by scDEED. **e–f** Dubious cell embeddings (**e**) and trustworthy cell embeddings (**f**) defined by scDEED at the perplexity 230.

## scDEED enhanced t-SNE visualization of the Hydra dataset

The Hydra scRNA-seq dataset was the first cell atlas of the adult Hydra polyp. The transcriptomes of 24,985 Hydra cells were sequenced by Drop-seq[29]. In the original study, the Hydra cells were visualized by t-SNE with the perplexity hyperparameter set to 40 (Fig. 2a, a reproduction of the original study's Fig. 1F in ref. 29).

We first applied scDEED to this original t-SNE visualization to detect dubious cell embeddings. The results had 4.77% of cells with

dubious embeddings (Figs. 2b) and 92.84% with trustworthy embeddings (Fig. 2c). The remaining cell embeddings, whose reliability scores were between the 5th and 95th percentiles of null reliability scores, were neither dubious nor trustworthy. Interestingly, most dubious cell embeddings appeared as small clusters, suggesting that these clusters might not represent disparate cell types. Moreover, there are three notable examples. First, the annotated *nematocytes* contained many cells with dubious embeddings (Supplementary Fig. S1a). We verified this finding by showing that these cells had gene expression profiles largely distinct from the other *nematocytes* with trustworthy embeddings (Supplementary Fig. S1b). Second, the annotated *neuron ec1* (*neuron ectodermal 1*) cells were divided into two non-neighboring clusters (Fig. 3a), but one cluster consisted of dubious cell embeddings (Fig. 2b). We examined the gene expression profiles of these two clusters and found the two clusters hardly distinguishable (Fig. 3c). Third, the annotated *male germline* cells had many dubious embeddings (Supplementary Fig. S2a). Checking the gene expression profiles of *male germline* cells with dubious embeddings, we found that these cells should belong to two clusters (Supplementary Fig. S2b) instead of the continuum shown in the original visualization. These results confirmed that scDEED effectively identified dubious cell embeddings worth further investigation.

Next, we used scDEED to optimize the t-SNE perplexity hyperparameter by minimizing the number of dubious cell embeddings (Fig. 2d); the optimized perplexity was 230 (Supplementary Fig. S3a, left). With this optimized perplexity, dubious cell embeddings decreased from 4.77% to 0.51% of all cell embeddings (Fig. 2e), and trustworthy cell embeddings increased from 92.84% to 99.1% (Fig. 2f). Notably, this optimized perplexity of 230 was close to Kobak and Berens' suggested t-SNE perplexity of 251, defined as the number of cells divided by 100[20]. Compared to this suggested perplexity, scDEED's optimized perplexity was data-driven and thus accounted for other data characteristics besides the cell number. In addition to finding the perplexity that minimized the number of dubious cell embeddings, we also considered the "kneedle" method[30] for finding the perplexity at the elbow point in the scatterplot of the number of dubious cell embeddings (y-axis) vs. the perplexity (x-axis). The resulting "kneedle" perplexity was 170 (Supplementary Fig. S3a, right), which gave similar cell embeddings to those output by scDEED's optimized perplexity of 230 (Supplementary Fig. S3b).

With scDEED's optimization, the t-SNE visualization (Fig. 2d) exhibited several key differences from the original t-SNE visualization (Fig. 2a). First, the two clusters of *neuron ec1* cells at the original perplexity of 40 (Fig. 3a) became one continuum at the perplexity of 230 optimized by scDEED (Fig. 3b). Using *ecEP_sc* (*ectodermal epithelial_single cell*) as a reference cell type, we observed a clearer picture: *ecEP_sc* was a close neighbor (with similar Euclidean distances in the 2D-embedding space) of the two separated *neuron ec1* clusters at the original perplexity (Fig. 3a); however, *ecEP_sc* became far away from the unified *neuron ec1* continuum at the perplexity optimized by scDEED (Fig. 3b). Gene expression profiles supported the visualization optimized by scDEED: the unified *neuron ec1* cells indeed had similar gene expression and a large distinction from *ecEP_sc* (Fig. 3c).

Second, at the original perplexity, the annotated *neuron ec1* cells and *neuron ec3* cells were distinct clusters that appeared similarly close to *ecEP_sc* cells (Fig. 3d); however, at the perplexity optimized by scDEED, *neuron ec1* cells and *neuron ec3* cells were unified as one large cloud far away from *ecEP_sc* (Fig. 3e). Again, the visualization optimized by scDEED was supported by those cells' gene expression profiles (Fig. 3f), which confirmed that *neuron ec1* cells and *neuron ec3* cells were indeed similar to each other and distinct from *ecEP_sc* cells.

Third, at the original perplexity, ectodermal neurons were shown in six clusters labeled as five subtypes: *neuron ec1*, *neuron ec2*, *neuron ec3*, *neuron ec4*, and *neuron ec5* (Fig. 3g). In contrast, at the perplexity optimized by scDEED, the five subtypes were unified (Fig. 3h). Using

*battery cell 2 (mp)* as a reference cell type, we found that *mp*'s location relative to the five subtypes changed drastically after scDEED optimized the perplexity: *mp* was closer to *neuron ec5* than the other four subtypes at the original perplexity (Fig. 3g), but *mp* became far away from all five subtypes at the optimized perplexity (Fig. 3h). Again, gene expression profiles supported the visualization optimized by scDEED (Fig. 3i). To further justify that the five subtypes should be unified, we calculated the ROGUE value, a cell cluster purity metric that ranges from 0 to 1 and whose higher value indicates a purer cluster[31], for every subtype and the unified cluster. The ROGUE values of the five subtypes had an average of 0.768 and a standard deviation of 0.055, while the ROGUE value of the unified cluster was 0.744. Hence, unifying the five subtypes did not significantly reduce the subtypes' purity and was thus deemed reasonable by us. These results confirmed that scDEED effectively optimized the t-SNE perplexity hyperparameter on this Hydra dataset.

## scDEED enhanced t-SNE visualization of the CAR-T dataset

The CAR-T scRNA-seq dataset contained 62,167 CD8+ chimeric antigen receptor modified-T (CAR-T) cells from 10 patients undergoing CD19 CAR-T immunotherapy[32]. In the original study, CAR-T cells were visualized by t-SNE with the perplexity of 30 (Fig. 4a, a reproduction of the original study's Fig. 5A with cell clusters highlighted as in ref. 32).

We first applied scDEED to this original visualization to detect dubious cell embeddings. The results had 1.81% of cells with dubious embeddings (Figs. 4b) and 83.99% with trustworthy embeddings (Fig. 4c). Most of the dubious cell embeddings were in *cluster 7*; at the original perplexity of 30, these dubious cell embeddings were not visually distinguishable from the trustworthy cell embeddings in *cluster 7* (Fig. 5a), suggesting the need for further investigation. To understand the differences between the dubious and trustworthy cell embeddings in *cluster 7*, we performed the differential gene expression analysis and found that the groups of cells indeed had distinct gene expression profiles (Fig. 5c; Supplementary Table 1). In particular, many ribosomal genes were more highly expressed in the trustworthily embedded cells, a phenomenon associated with higher transcriptional activities[33,34]. This observation was consistent with the fact the trustworthily embedded cells were more enriched with CAR-T cells from the early stimulation stage than the dubiously embedded cells (48.05% of trustworthily embedded cells and 28.51% of dubiously embedded cells were from the early simulation stage; *p* value = 3.304e-12). Since the early-stage CAR-T cells right after simulation were expected to have higher transcriptional activities than the other CAR-T cells, it is reasonable that the trustworthily embedded cells had higher ribosomal gene expression than the dubiously embedded cells.

Next, we used scDEED to optimize the t-SNE perplexity hyperparameter by minimizing the number of dubious cell embeddings; the resulting perplexity was 750 (Supplementary Fig. S4a, left). With this optimized perplexity, dubious cell embeddings decreased from 1.81% to 0.09% of all cell embeddings (Fig. 4e), and trustworthy cell embeddings increased from 83.99% to 95.87% (Fig. 4f). The "kneedle" method resulted in a perplexity of 170 (Supplementary Fig. S4a, right), which gave similar cell embeddings to those of the optimized perplexity 750 (Supplementary Fig. S4b).

With the perplexity of 750 optimized by scDEED, the t-SNE visualization (Fig. 4d) exhibited several key differences from the original t-SNE visualization (Fig. 4a). First, the dubiously embedded and trustworthily embedded cells in *cluster 7* were no longer in one cluster (Fig. 5a) but separated far apart at the perplexity optimized by scDEED (Fig. 5b). Taking *cluster 5* and *cluster 14* as references, we observed that the dubiously embedded cells in *cluster 7* became close to *cluster 14*, while the trustworthily embedded cells in *cluster 7* became close to *cluster 5* (Supplementary Fig. S5). We validated this visualization

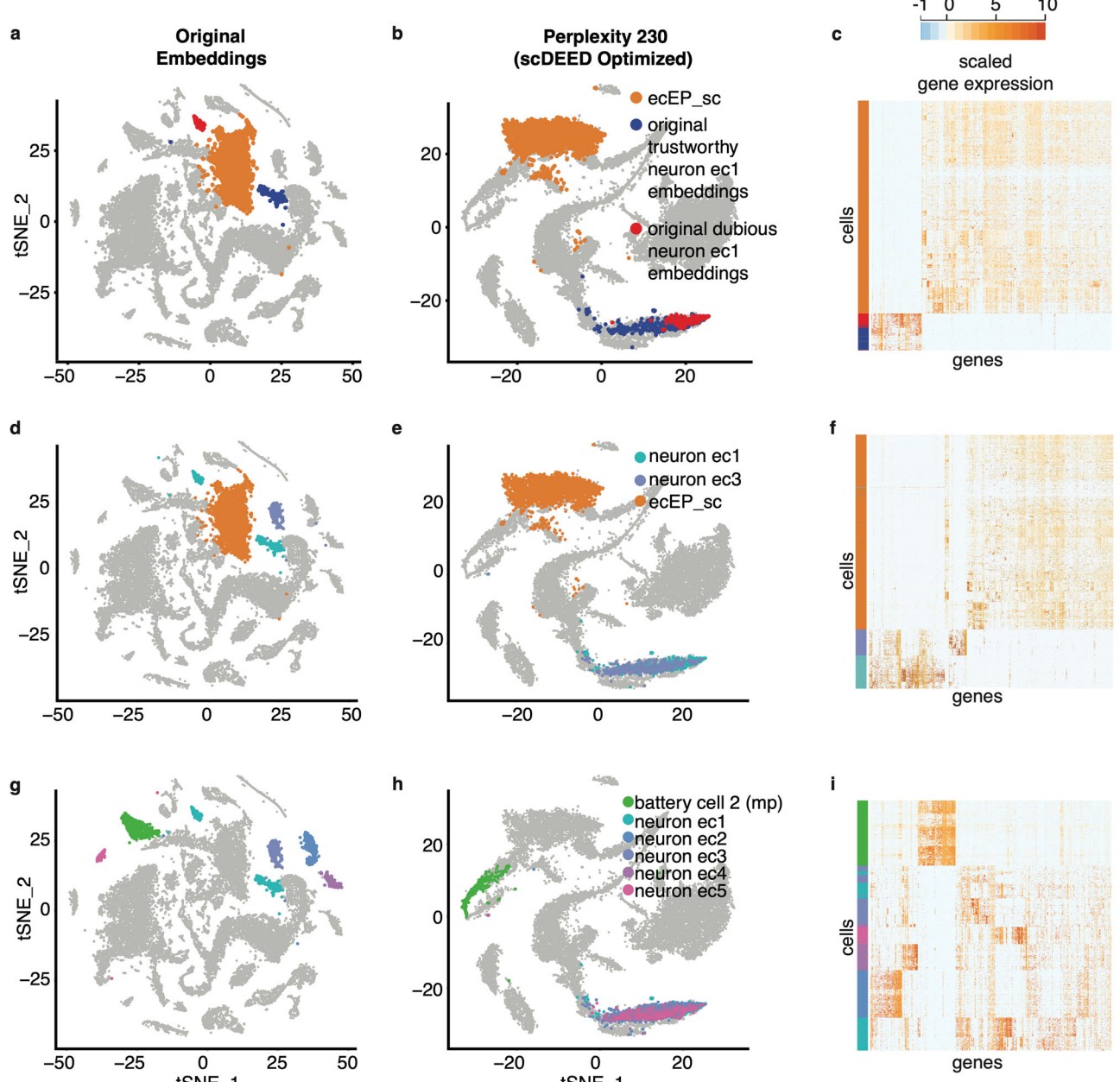

**Fig. 3 | Evaluation of t-SNE embeddings optimized by scDEED on the Hydra dataset. a–b** Comparative t-SNE plots with the *ecEP_sc (ectodermal epithelial_single cell)*, *trustworthy cell embeddings in neuron ec1*, and *dubious cell embeddings in neuron ec1* highlighted, at the original perplexity 40 (**a**) and the perplexity 230 optimized by scDEED (**b**). **c** Gene expression heatmap of the highlighted cells in (**a**) and (**b**), where the cells are ordered by the default hierarchical clustering found by the R function heatmap.2(). **d–e** Comparative t-SNE plots with the *neuron ec1*, *neuron ec3*, and *ecEP_sc* cells highlighted. At the original perplexity 40 (**d**), the *neuron ec1* cells are in two separate clusters and have similarly short distances as the *neuron ec3* cells have to the *ecEP_sc* cells; at the optimized perplexity 230 (**e**), the *neuron ec1* and *neuron ec3* cells are unified as one cluster far away from the *ecEP_sc*

cells. **f** Gene expression heatmaps of the highlighted cells in (**d**) and (**e**), where the cells are ordered by the default hierarchical clustering found by the R function heatmap.2(). **g–h** Comparative t-SNE plots with the *neuron ec1*, *neuron ec2*, *neuron ec3*, *neuron ec4*, *neuron ec5*, and *battery cell 2 (mp)* cells highlighted. At the original perplexity 40 (**g**), the *neuron ec1*, *neuron ec2*, *neuron ec3*, *neuron ec4*, and *neuron ec5* cells are in distinct clusters surrounding the *battery cell 2 (mp)* cells; at the optimized perplexity 210 (**h**), the five *neuron ec* clusters are unified as one cluster far away from the *battery cell 2 (mp)* cells. **i** Gene expression heatmap of the highlighted cells in (**g**) and (**h**), where the cells are ordered by the default hierarchical clustering found by the R function heatmap.2().

using the cell trajectory reconstruction method STREAM[35] (Supplementary Fig. S5b): the reconstructed cell trajectory had the dubiously embedded *cluster 7* cells in one branch with the *cluster 14* cells (branch S6), while the trustworthily embedded *cluster 7* cells were in another branch with the *cluster 5* cells (branch S4).

Second, at the original perplexity, *cluster 7* was between *cluster 14* and *cluster 6*; in particular, *cluster 7* was next to *cluster 14* and at some distance from *cluster 6* (Fig. 5d). However, this pattern changed after

we optimized the perplexity. In particular, at the perplexity optimized by scDEED, most of the dubiously embedded *cluster 7* cells were in one small cluster next to *cluster 14*, but the rest of *cluster 7* cells (including the trustworthily embedded *cluster 7* cells) were no longer between *cluster 14* and *cluster 6* (Fig. 5e). This optimized visualization was consistent with the gene expression profiles (Fig. 5f).

Third, the four clusters under the stage IP (no stimulation), i.e., *clusters 3, 4, 9, and 11* (Fig. 5A in the original study[32]), had little overlap

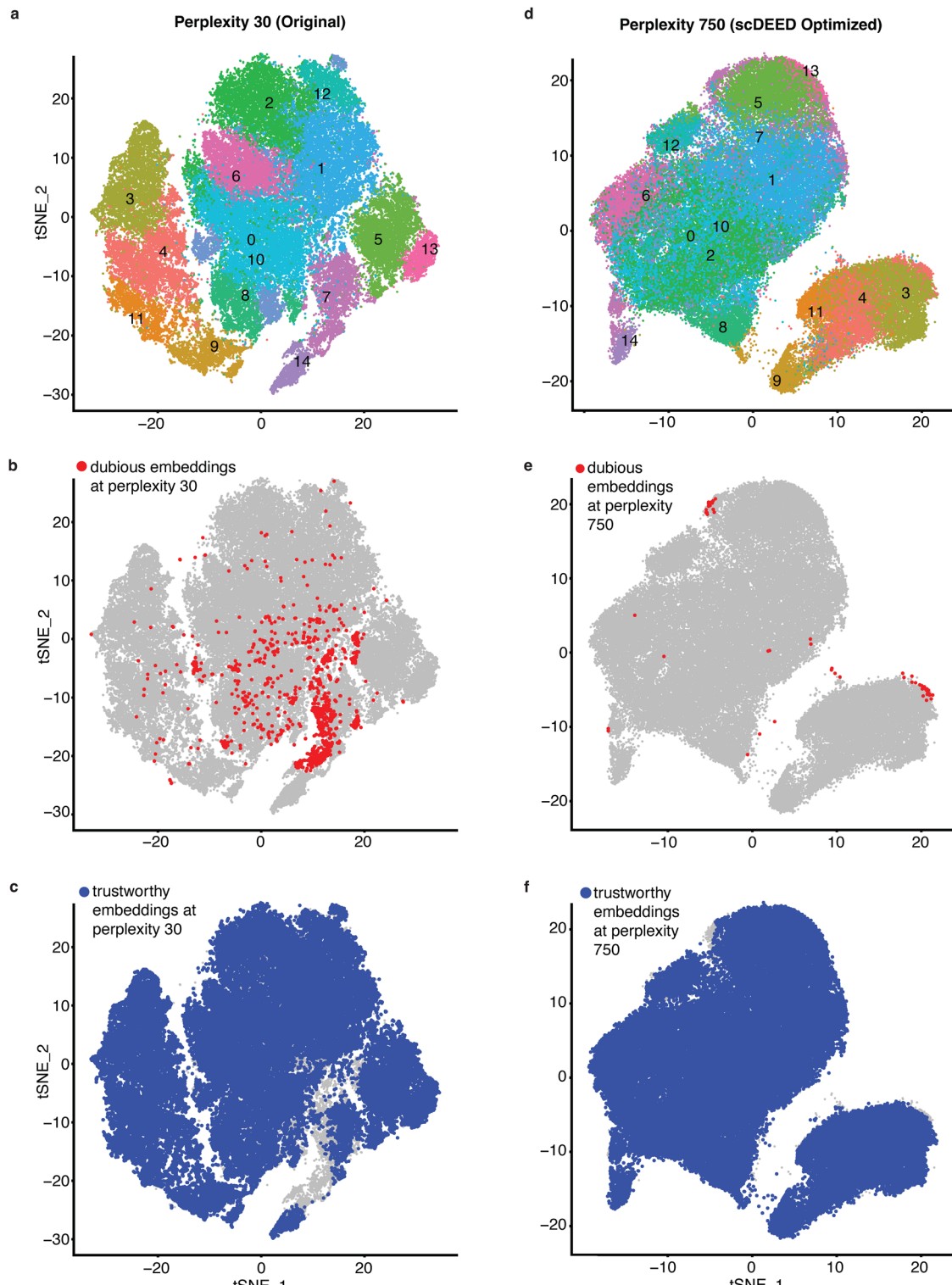

**Fig. 4 | Original t-SNE embeddings and t-SNE embeddings optimized by scDEED on the CAR-T dataset. a** t-SNE plot of the CAR-T dataset at the study's original perplexity 30. Dubious embeddings (**b**) and trustworthy embeddings (**c**) defined by scDEED at the original perplexity 30. **d** t-SNE plot of the CAR-T dataset at the perplexity 750 optimized by scDEED. Dubious embeddings (**e**) and trustworthy embeddings (**f**) defined by scDEED at the optimized perplexity 750.

at the original perplexity (Fig. 5g). Nevertheless, at the perplexity optimized by scDEED, *cluster 9* stayed distinct, while the other three clusters (*clusters 3, 4,* and *11*) had large overlaps (Fig. 5h). We found supportive evidence in the gene expression profiles, showing that *cluster 9* was distinct from the other three clusters (Fig. 5i). To confirm this result quantitatively, we computed the ROGUE values for *cluster 3,*

the combination of *clusters 3 and 4,* and the combination of *clusters 3, 4,* and *11,* obtaining 0.887, 0.885, and 0.886, respectively. These ROGUE values suggested no clear separation among the three clusters. Moreover, the ROGUE value dropped to 0.832 after we combined *clusters 3, 4, 9,* and *11,* indicating that *cluster 9* had distinct gene expression profiles from those of *clusters 3, 4,* and *11.* Together, the

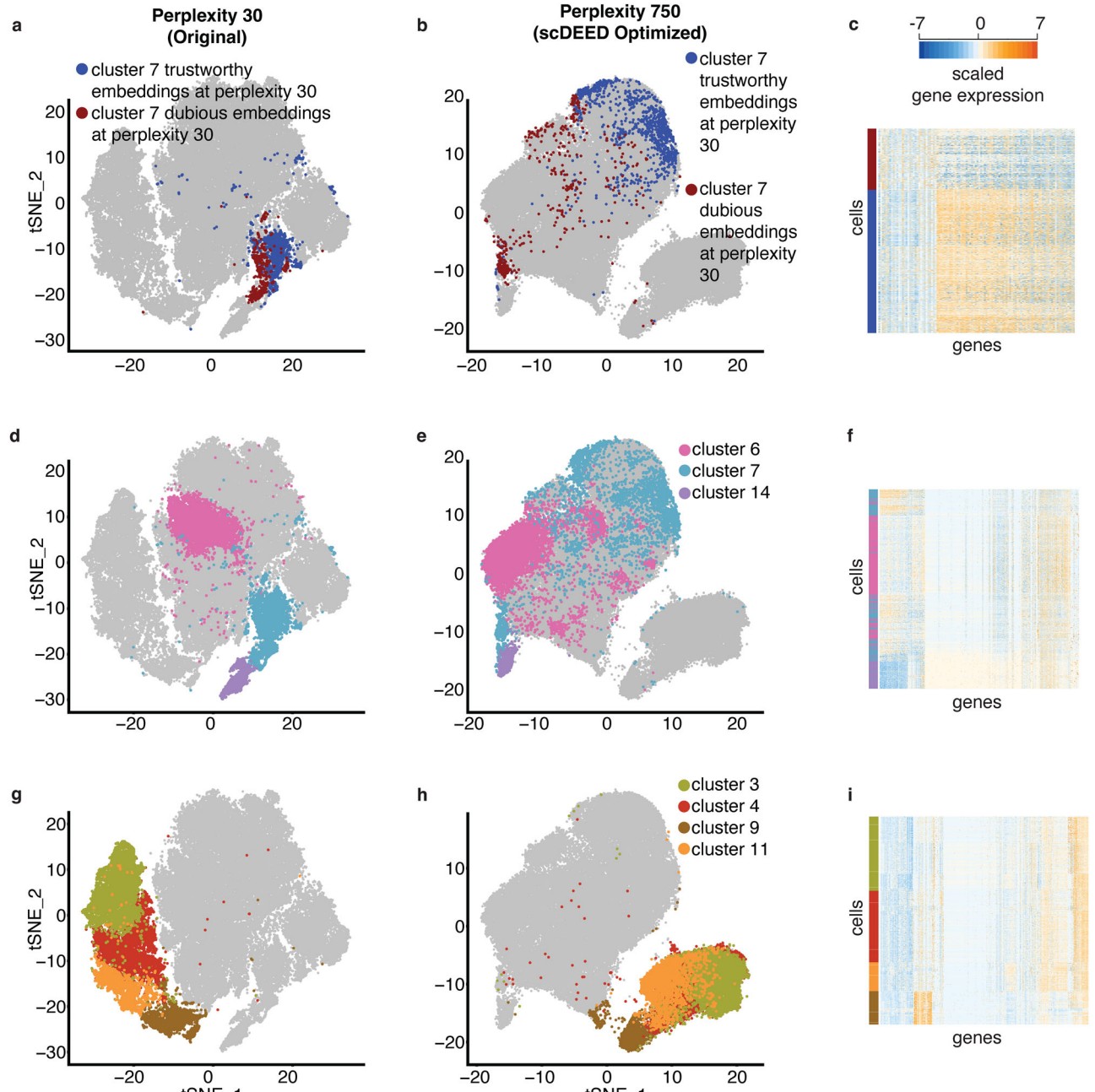

**Fig. 5 | Evaluation of cluster locations at the original t-SNE perplexity and the perplexity optimized by scDEED on the CAR-T dataset.** Comparative t-SNE plots of *cluster 7*'s dubious and trustworthy cell embeddings at the original perplexity 30 (**a**) and the perplexity 750 optimized by scDEED (**b**). **c** Gene expression heatmap of highlighted cells in (**a**) and (**b**). Comparative t-SNE plots of clusters *6, 7,* and *14* at the perplexity 30 in the original study (**d**) and the perplexity 750 optimized by scDEED (**e**). We have recolored *cluster 7* for better visualization in (**d**–**f**). **f** Gene expression heatmap of the highlighted cells in (**d**) and (**e**), where cells are ordered by the default hierarchical clustering found by the R function `heatmap.2()`. Comparative t-SNE plots of *clusters 3, 4, 9, and 11* at the perplexity 30 in the original study (**g**) and the perplexity 750 optimized by scDEED (**h**). We have recolored the clusters for better visualization in (**g**–**i**). **i** Gene expression heatmap of the highlighted cells in (**g**) and (**h**).

ROGUE values were consistent with the t-SNE visualization at the perplexity optimized by scDEED.

Fourth, at the original perplexity, *cluster 8* and *cluster 9* appeared to have similar distances from *cluster 14* (Supplementary Fig. S6a left). However, *cluster 8* and *cluster 9*'s relative locations changed after we optimized the perplexity, with *cluster 8* standing between *cluster 14* and *cluster 9* (Supplementary Fig. S6a right). Again, their gene expression profiles and the corresponding hierarchical clustering of cells supported the t-SNE visualization under the optimized perplexity (Supplementary Fig. S6b).

## scDEED enhanced UMAP visualization of the Alveolar dataset

The Alveolar dataset was collected to learn the cellular dynamics during the regeneration process after bleomycin-induced lung injury[36]. Measured by Dropseq, the Alveolar dataset contained 29,297 cells from 28 mice, with about 1000 cells per mouse. In the original study, cells were visualized by UMAP with hyperparameters min.dist = 0.3 and n.neighbors = 10 (Fig. 6a, a reproduction of the original study's Fig. 1a[36] with the annotated cell clusters highlighted).

We applied scDEED to this original visualization to detect dubious cell embeddings. The results show that 3.14% of cells had dubious

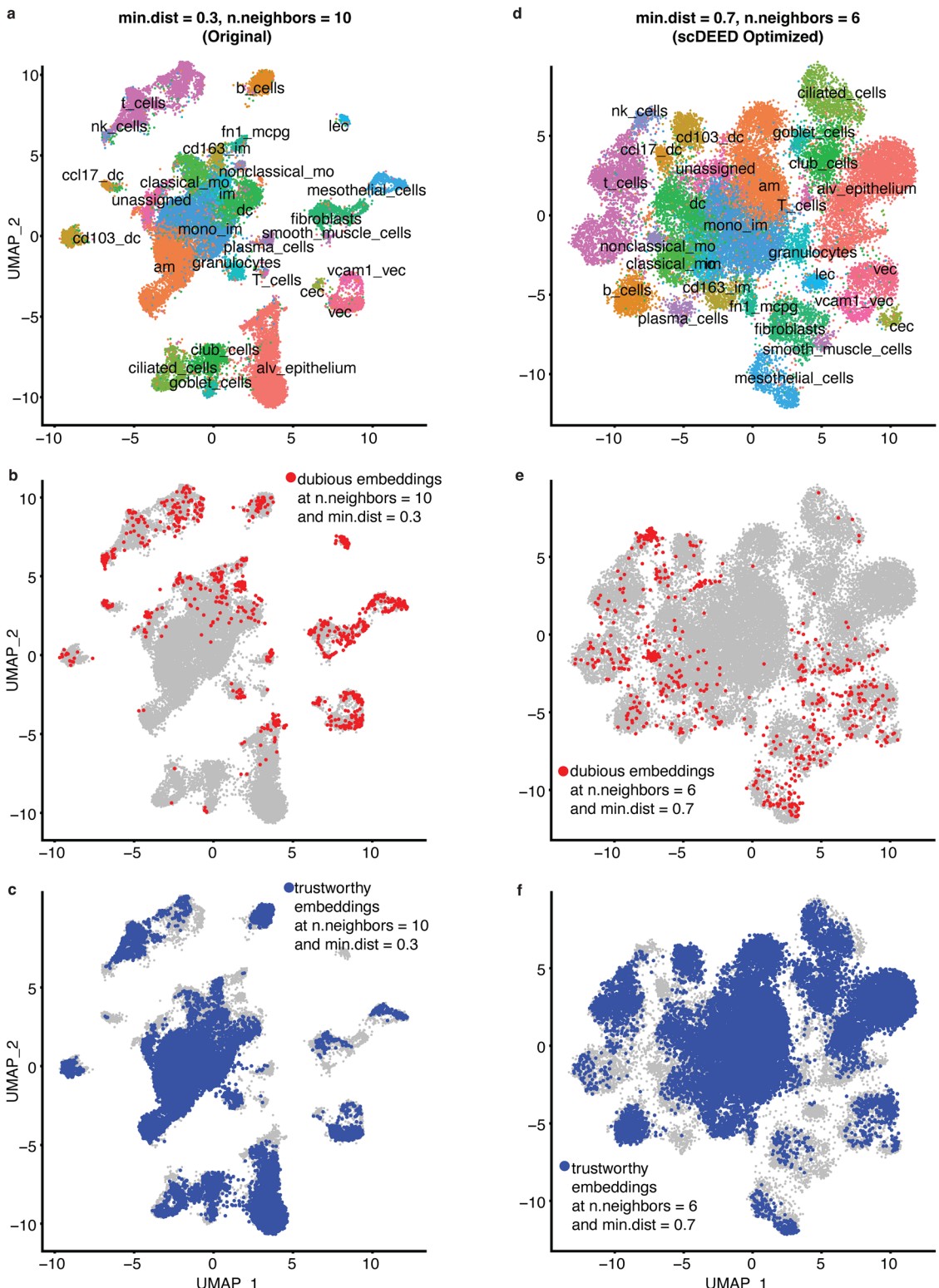

**Fig. 6 | Original UMAP embeddings and UMAP embeddings optimized by scDEED on the Alveolar dataset. a** UMAP plot of the Alveolar dataset at the study's original hyperparameters, min.dist = 0.3 and n.neighbors = 10. Dubious embeddings (**b**) and trustworthy embeddings (**c**) defined by scDEED at the original

hyperparameters. **d** UMAP plot of the Alveolar dataset at the hyperparameters jointly optimized by scDEED, min.dist = 0.7 and n.neighbors = 6. Dubious embeddings (**e**) and trustworthy embeddings (**f**) defined by scDEED at the optimized hyperparameters.

embeddings (Figs. 6b), and 50.71% had trustworthy embeddings (Fig. 6c). Unlike t-SNE, UMAP has two hyperparameters: min.dist and n.neighbors, which jointly determine the 2D cell embeddings. The scDEED R package allows users to optimize min.dist and n.neighbors

marginally or jointly, and the goal is to minimize the number of dubious embeddings. In marginal optimization, we applied scDEED to optimize min.dist or n.neighbors by fixing the other hyperparameter at the original value. Hence, marginal optimization considers fewer

hyperparameter combinations and is thus more computationally efficient than joint optimization.

From marginal optimization, we obtained n.neighbors = 5 (with min.dist = 0.3) and min.dist = 0.6 (with n.neighbors = 10) (Supplementary Fig. S7). In particular, the first hyperparameter set (min.dist = 0.3 and n.neighbors = 5) reduced the dubious embeddings to 2.28% of cells and increased the trustworthy embeddings to 56.73%; the second hyperparameter set (min.dist = 0.6 and n.neighbors = 10) reduced the dubious embeddings to 2.45% and increased the trustworthy embeddings to 54.75%. When we jointly optimized the two hyperparameters, we obtained min.dist = 0.7 and n.neighbors = 6 (Fig. 6d), which further reduced the dubious embeddings to 2.04% (Fig. 6e) and increased the trustworthy embeddings to 58.84% (Fig. 6f).

Among the cell clusters that contained many dubious cell embeddings, we focused on the relative locations of the following clusters: *lec*, *b_cells*, *vcam1_vec*, *vec*, and *t_cells*. We noted two key differences between the original UMAP visualization (Fig. 7a, d) and the three optimized visualizations (joint hyperparameter optimization in Fig. 7b, e; marginal hyperparameter optimizations in Supplementary Fig. S8a–b); note that in the three optimized visualizations, the relative cell type locations are more similar to each other than to those in the original visualization.

First, under the original hyperparameter setting (min.dist = 0.3 and n.neighbors = 10), the *lec* cluster was close to the *b_cells* cluster and far away from the *vcam1_vec* and *vec* clusters (Fig. 7a). In contrast, under the three optimized hyperparameter settings, *lec* was no longer adjacent to *b_cells* but became close to *vcam1_vec* and *vec*, with *b_cells* lying far away (Fig. 7b and Supplementary Fig. S8a). We found supporting evidence in the gene expression profiles, which showed that the *b_cells* cluster was distinct from the *lec*, *vcam1_vec*, and *vec* clusters (Fig. 7c).

Second, under the original hyperparameter setting, the *b_cells* cluster lay between the *t_cells* and *lec* clusters at approximately equal distances (Fig. 7d). In contrast, under the three optimized hyperparameter settings, the *b_cells* and *t_cells* clusters became close to each other but far away from the *lec* cluster (joint hyperparameter optimization in Fig. 7e; marginal hyperparameter optimizations in Supplementary Fig. S8b). Again, to evaluate the relative locations of the *b_cells*, *t_cells*, and *lec* clusters, we examined the gene expression profiles. Figure 7f shows that *lec* was the most distinctive among the three clusters, thus supporting the optimized visualizations found by scDEED.

### scDEED enhanced UMAP visualization of the Samusik dataset

The Samusik dataset was from a mass cytometry study of bone marrow hematopoiesis. It contained more than 86,000 cells, 38 cell phenotype features, and 24 annotated cell types[37]. In the original study[37], cells were visualized by UMAP with hyperparameters min.dist = 0.2 and n.neighbors = 15 (Fig. 8a, a reproduction of the original study's Fig. 2a[37]).

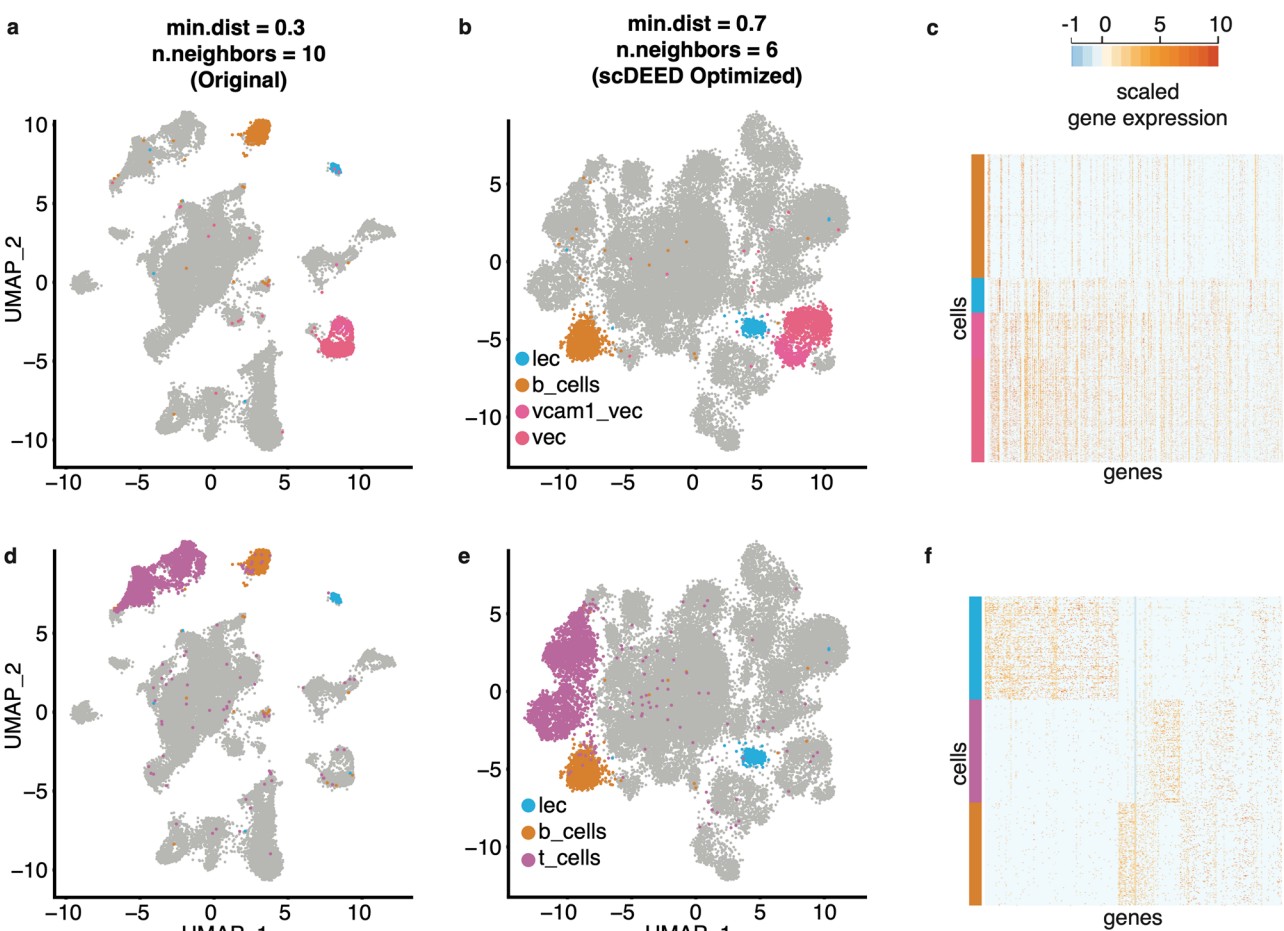

**Fig. 7 | Evaluation of cluster locations at the original UMAP hyperparameters and the hyperparameters jointly optimized by scDEED on the Alveolar dataset.** Comparative UMAP plots of the Alveolar dataset with the *lec*, *b_cells*, *vcam1_vec*, and *vec* cells highlighted at the original hyperparameters of min.dist = 0.3 and n.neighbors = 10 (**a**) and the hyperparameters of min.dist = 0.7 and n.neighbors = 6 jointly optimized by scDEED (**b**). **c** Gene expression heatmap of the highlighted cells in (**a**) and (**b**). Comparative UMAP plots of the Alveolar dataset with the *lec*, *b_cells*, and *t_cells* cells highlighted at the original hyperparameters (**d**) and the hyperparameters jointly optimized by scDEED (**e**). **f** Gene expression heatmap of the highlighted cells in (**d**) and (**e**). Note we randomly downsampled *b_cells* and *t_cells* (from 911 and 2709 cells, respectively) so that each cluster has 256 cells (same as the number of cells in *lec*) to make a visually informative heatmap.

We first applied scDEED to this original visualization, finding 1.13% dubious cell embeddings (Fig. 8b) and 98.74% trustworthy cell embeddings (Fig. 8c). Next, we applied scDEED to optimize min.dist and n.neighbors marginally by minimizing the number of dubious cell embeddings. From this marginal optimization, we obtained

n.neighbors = 160 (with min.dist = 0.2) and min.dist = 0.05 (with n.neighbors = 15) (Supplementary Fig. S9). In particular, the first hyperparameter set (min.dist = 0.2 and n.neighbors = 160) reduced the dubious embeddings to 0.68% and increased the trustworthy embeddings to 99.19%; the second hyperparameter set (min.dist = 0.05 and

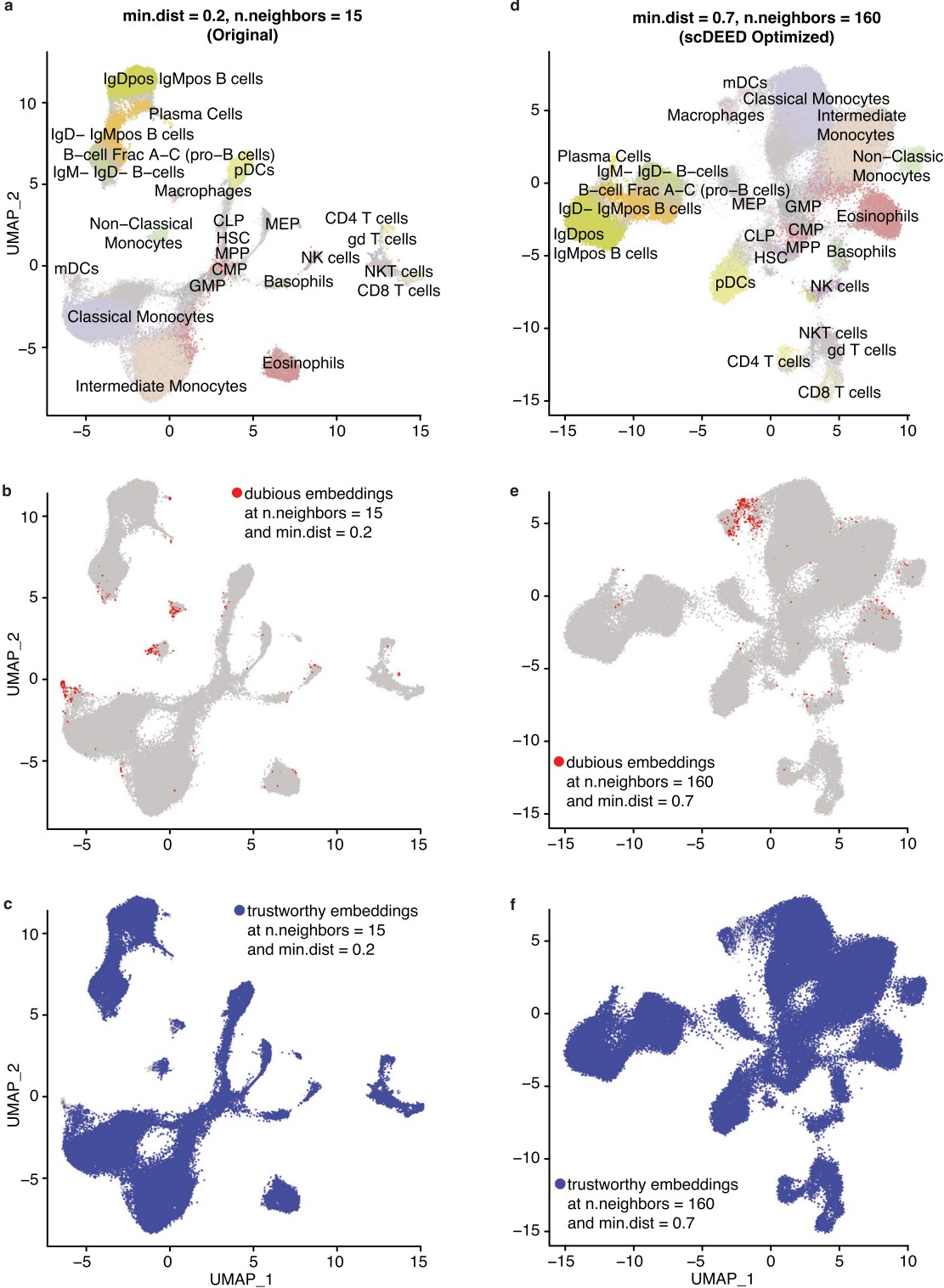

Fig. 8 | Original UMAP embeddings and UMAP embeddings optimized by scDEED on the Samusik dataset. a UMAP plot of the Samusik dataset at the study's original hyperparameters, min.dist = 0.2 and n.neighbors = 15. Dubious embeddings (b) and trustworthy embeddings (c) defined by scDEED at the original hyperparameters. d UMAP plot of the Samusik dataset with the hyperparameters jointly optimized by scDEED, min.dist = 0.7 and n.neighbors = 160. Dubious embeddings (e) and trustworthy embeddings (f) defined by scDEED at the optimized hyperparameters.

n.neighbors = 15) reduced the dubious embeddings to 0.837% and increased the trustworthy embeddings to 99.028%. When we jointly optimized the two hyperparameters, we obtained min.dist = 0.7 and n.neighbors = 160 (Fig. 8d). This hyperparameter set further reduced the dubious embeddings to 0.641% (Fig. 8e) and increased the trustworthy embeddings to 99.269% (Fig. 8f). As expected, the joint optimization achieved the lowest percentage of dubious cell embeddings and the highest percentage of trustworthy cell embeddings. Meanwhile, the joint optimization and the marginal optimization of n.neighbors shared the same n.neighbors = 160 and similar visualizations (joint hyperparameter optimization in Fig. 8d; marginal hyperparameter optimizations in Supplementary Fig. S9b)

Among the cell types that contained dubious cell embeddings, we focused on the relative locations of the following cell types: *Non-Classical Monocytes* (*ncm*), *Intermediate Monocytes, Basophils, Plasmacytoid Dendritic Cells (pDCs), Myeloid Dendritic Cells (mDCs)*, and *Macrophages*. Notably, the joint optimization and the marginal optimization of n.neighbors (Fig. 8d and Supplementary Fig. S9b left) exhibited two key differences from the original UMAP visualization (Fig. 8a).

First, under the original hyperparameter setting (min.dist = 0.2 and n.neighbors = 15), *Macrophages* lay between *pDCs* and *ncm* with

approximately equal distances and were far away from *mDCs* (Fig. 9a). In contrast, under the marginally optimized n.neighbors setting (min.dist = 0.2 and n.neighbors = 160) and the jointly optimized hyperparameter setting (min.dist = 0.7 and n.neighbors = 160), which had the two lowest percentages of dubious embeddings, *Macrophages* became adjacent to *mDCs* but farther away from *ncm* and *pDCs* (Fig. 9b). The gene expression profiles confirmed that *Macrophages* were more similar to *mDCs* than *ncm* and *pDCs* (Fig. 9c).

Second, under the original hyperparameter setting, looking at pairwise distances among the three cell types—*Plasma Cells, ncm*, and *NK cells*, we found similar distances between *Plasma Cells* and *ncm*, and between *ncm* and *NK cells*, and the distance between *Plasma Cells* and *NK cells* was the largest (Fig. 9d). In contrast, under the jointly optimized hyperparameter setting (min.dist = 0.7 and n.neighbors = 160), the distance between *ncm* and *NK cells* became the smallest among the three pairwise distances, with the other two distances becoming similar to each other (Fig. 9e). To evaluate the relative locations of these three cell types, we calculated the cell-to-cell Euclidean distances in the 38-dimensional feature space (before UMAP embedding) between every two cell types, confirming that *NK cells* and *ncm* were closer to each other than to *Plasma Cells* (Fig. 9f).

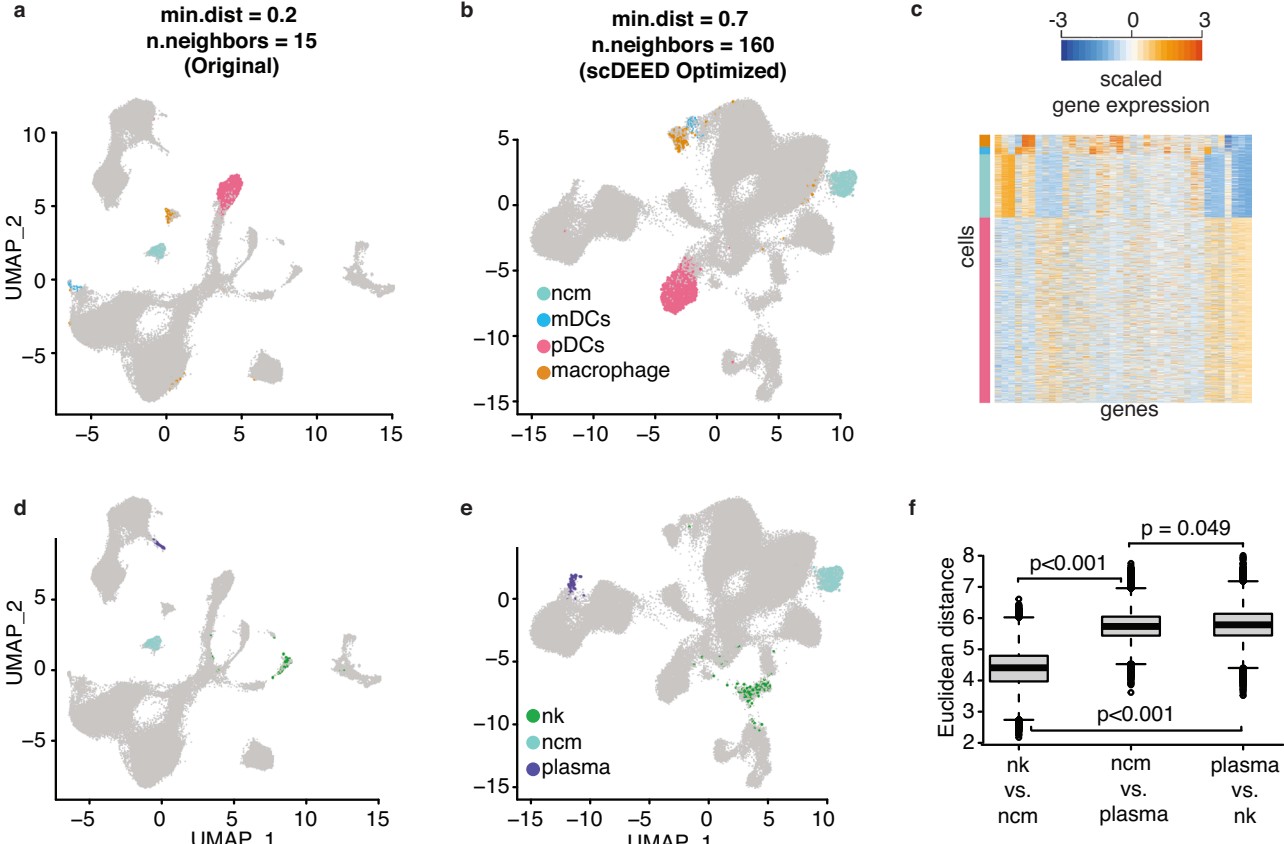

**Fig. 9 | Evaluation of cluster locations at the original UMAP hyperparameters and the hyperparameters optimized by scDEED on the Samusik dataset.** Comparative UMAP plots of the Samusik dataset with the *non-classical monocytes (ncm)*, *mDCs*, *pDCs*, and *macrophages* highlighted at the original hyperparameters of min.dist = 0.2 and n.neighbors = 15 (**a**) and the hyperparameters of min.dist = 0.7 and n.neighbors = 160 jointly optimized by scDEED (**b**). **c** Gene expression heatmap of the highlighted cells in (**a**) and (**b**). Comparative UMAP plots of the Samusik dataset with the *NK cells (nk)*, *ncm*, and *plasma cells* highlighted at the original hyperparameters (**d**) and the hyperparameters optimized by scDEED (**e**). **f** Comparison of distances between *ncm*, *nk*, and *plasma* cells. The box center lines, bounds, and whiskers denote the medians, first and third quartiles, and minimum

and maximum values within 1.5 × the interquartile range of the box limits, respectively. The two-sample t statistic p-values for between-boxplot comparisons are presented, with the null distribution computed based on 1000 random partitions of the cells in the three types by preserving the three cell type sizes (the theoretical t distribution should not be used because the distances are not independent). The two-sample *t* statistics are as follows: (*nk* vs. *ncm*, *n* = 192,024 pairs) vs (*ncm* vs. *plasma*, *n* = 119,888 pairs) = −728.580 (*p* < 0.001), (*nk* vs. *ncm*, *n* = 192,024 pairs) vs (*plasma* vs. *nk*, *n* = 22,302 pairs) = −370.042 (*p* < 0.001), (*ncm* vs. *plasma*, *n* = 119,888 pairs) vs (*plasma* vs. *nk*, *n* = 22,302 pairs) = −13.094 (*p* = 0.049). **f** confirms that (**e**) better preserves the three clusters' relative distances than (**d**) does. Source data are provided as a Source Data file uploaded on Zenodo.

Note that the above two differences were not as apparent in the embeddings obtained from marginally optimizing the hyperparameter min.dist (min.dist = 0.05 and n.neighbors = 15). A possible reason was that this hyperparameter set had the highest percentage of dubious embeddings and the lowest percentage of trustworthy embeddings among the three optimized hyperparameter sets.

## scDEED improved the consistency between t-SNE and UMAP
The Human PBMC dataset was collected to compare multiple scRNA-seq technologies[38]. It contained 31,021 cells with cell type labels, and the gene expression levels were in the unit of log-transformed UMI count per 10,000. We accessed the dataset "pbmcsca.SeuratData" in the R package "SeuratData."

We considered three scRNA-seq technologies that measured more than 500 cells in the dataset: Dropseq, inDrops, and SeqWell. For each technology, we visualized its measured cells using t-SNE and UMAP at the default hyperparameters or the hyperparameters optimized by scDEED (Fig. 10). We observed that the optimized hyperparameters led to more consistent relative distances among the cell

types, both across scRNA-seq technologies and between t-SNE and UMAP. Specifically, we had the following three observations.

First, when used to visualize the Dropseq data at the default hyperparameters, t-SNE and UMAP showed different relative distances among four cell types: t-SNE ordered the cell types *Cytotoxic T cell*, *CD4 + T cell*, *CD14+ monocyte*, and *B cell* clockwise (Fig. 10a, left), but UMAP switched the order of *CD14+ monocyte* and *B cell* (Fig. 10a, right). In contrast, at the hyperparameters optimized by scDEED, t-SNE and UMAP had the same counterclockwise order of the four cell types: *Cytotoxic T cell*, *CD4 + T cell*, *B cell*, and *CD14+ monocyte* (Fig. 10b). That is, the hyperparameters optimized by scDEED improved the consistency between t-SNE and UMAP.

Second, when used to visualize the inDrops data at the default hyperparameters, t-SNE ordered *Cytotoxic T cell*, *CD4 + T cell*, *B cell*, and *CD14+ monocyte* counterclockwise, while UMAP ordered the same four cell types clockwise (Fig. 10c). In contrast, at the hyperparameters optimized by scDEED, both t-SNE and UMAP ordered the four cell types clockwise (Fig. 10d). Notably, this clockwise order was consistent with the counterclockwise order of the four cell types in the t-SNE and

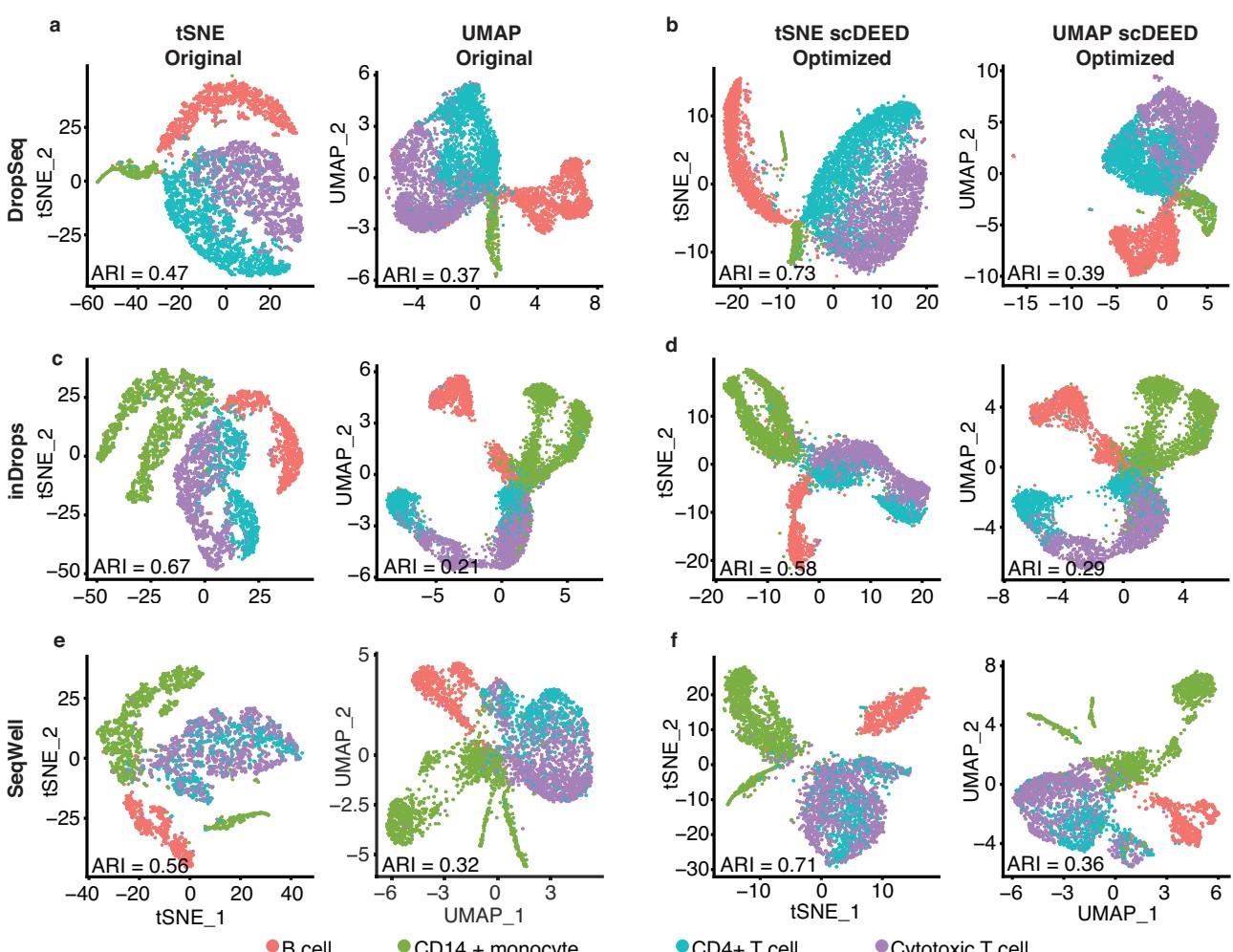

**Fig. 10 | Original t-SNE and UMAP embeddings and embeddings optimized by scDEED on the Human PBMC dataset. a** t-SNE and UMAP plots for the DropSeq dataset at the original hyperparameters, perplexity = 30 (left) and min.dist = 0.3 and n.neighbors = 30 (right). **b** t-SNE and UMAP plots for the Dropseq dataset at the hyperparameters optimized by scDEED, perplexity = 290 (left) and min.dist = 0.5 and n.neighbors = 5 (right). **c** t-SNE and UMAP plots for the inDrops dataset at the original hyperparameters, perplexity = 30 (left) and min.dist = 0.3 and n.neighbors = 30 (right). **d** t-SNE and UMAP plots for the inDrops dataset at the hyperparameters optimized by scDEED, perplexity = 320 (left) and min.dist = 0.5 and

n.neighbors = 80 (right). **e** t-SNE and UMAP plots for the SeqWell dataset at the original hyperparameters, perplexity = 30 (left) and min.dist = 0.3 and n.neighbors = 30 (right). **f** t-SNE and UMAP plots for the SeqWell dataset at the hyperparameters optimized by scDEED, perplexity = 140 (left) and min.dist = 0.2 and n.neighbors = 7 (right). Applying spectral clustering to identify cell clusters of the same number as the cell types in each set of embeddings, we found that in five out of six cases (three scRNA-seq technologies with t-SNE and UMAP embeddings), the optimized embeddings led to a higher adjusted Rand index (ARI) than the original embeddings, suggesting that the optimized embeddings better represented the cell types.

UMAP visualizations of the Dropseq data. Hence, the relative distances of the four cell types became consistent between Dropseq and inDrops after scDEED's optimization.

Third, when used to visualize the Seqwell data at the default perplexity hyperparameter, t-SNE separated the *CD14+ monocyte* cell type into two clusters, with the two cell types *Cytotoxic T cell* and *CD4 + T cell* in between (Fig. 10e, left); however, the two *CD14+ monocyte* clusters became attached at the perplexity optimized by scDEED (Fig. 10f, left). The optimized t-SNE and UMAP visualizations had a consistent counterclockwise order of the four cell types *Cytotoxic T cell*, *CD4 + T cell*, *B cell*, and *CD14+ monocyte* (Fig. 10f). Hence, the relative distances of the four cell types became consistent across the three scRNA-seq technologies after scDEED's optimization.

Finally, we verified that the 2D cell embeddings became better aligned with the cell types after scDEED's optimization. Towards this goal, we applied spectral clustering to the original embeddings and scDEED's optimized embeddings to identify cell clusters and checked the agreement with the cell types by calculating the adjusted Rand index (ARI)[39]. We used spectral clustering because it is capable of identifying clusters of non-spherical shapes. As a measure of the agreement between cell cluster labels and cell type labels, ARI adjusts for the agreement due to random chance. We found that in five out of six cases (three scRNA-seq technologies with t-SNE and UMAP embeddings), the optimized embeddings led to higher ARIs than the original embeddings (Fig. 10), suggesting that the optimized embeddings better represented the cell types.

### scDEED improved t-SNE visualization of RNA velocities
We performed RNA velocity analysis[40] on a dentate gyrus dataset[41]. RNA velocity vectors were calculated in the high-dimensional space, and a 2D vector field was used for visualization. The visualized 2D vectors were calculated based on small neighborhoods of cells defined in the 2D embedding space. Hence, 2D embedding affected velocity visualization.

Supplementary Fig. S10a shows a subset of the cells in the t-SNE visualization of the original study[41], and Supplementary Fig. S10c shows the same cells in the t-SNE visualization under the perplexity optimized by scDEED. Of particular note was the trajectory from *nIPC* to neuronal subtypes 1 and 2 (*Neuro1* and *Neuro2*), then to *immature granules*, and ending at *mature granules*[41]. This velocity trajectory became more evident in the t-SNE visualization at the perplexity optimized by scDEED (Supplementary Fig. S10d) than at the original perplexity (Supplementary Fig. S10b). Aside from the cells in the trajectory, the *mature granules* continued to have velocities close to zero, consistent with the fact that *mature granules* are at the end of differentiation. Hence, scDEED's optimized perplexity improved the t-SNE visualization of RNA velocities.

### scDEED outperformed EMBEDR in optimizing t-SNE and UMAP
We benchmarked scDEED against EMBEDR, a method designed based on the t-SNE loss function, on three datasets: (1) the Hydra dataset (used in the "Results" section "scDEED enhanced t-SNE visualization of the Hydra dataset"), (2) the Tabula Muris Consortium marrow dataset[42] (used in the EMBEDR paper[25]), and (3) the synthetic data generated by scDesign3[43]. We compared scDEED with EMBEDR in three aspects: detection of dubious embeddings at the default t-SNE perplexity, optimization of the t-SNE perplexity, and computational time.

On the Hydra dataset, at the original t-SNE perplexity of 40, unlike scDEED (Fig. 2b; Supplementary Fig. S11a), EMBEDR found most of the cell embeddings dubious (Supplementary Fig. S11b). We argue that the EMBEDR result was unlikely reasonable because if most of the cell embeddings were dubious, then the t-SNE visualization used in the Hydra study[29] would be meaningless; however, this was not the case.

Besides the number of dubious embeddings, scDEED and EMBEDR had two notable differences in their detection results on the Hydra dataset. First, among the *nematocytes*, the cells with dubious embeddings are more distinct from those with trustworthy embeddings in the scDEED result (Supplementary Fig. S11c and e; Supplementary Fig. S1) than in the EMBEDR result (Supplementary Fig. S11d and f). To quantify the difference between the scDEED and EMBEDR results, we calculated the pairwise distances between dubious-embedding cells and trustworthy-embedding cells in the scDEED and EMBEDR results separately; then we compared the two sets of pairwise distances using the one-sided Wilcoxon rank-sum test and obtained a *p* value less than $10^{-16}$, suggesting that the dubious embeddings and trustworthy embeddings found by scDEED were more distinct and thus more reasonable. Second, for the *neuron ec1* cells divided into two non-neighboring clusters (Fig. 3a), scDEED identified one cluster as dubious and the other cluster as trustworthy (Supplementary Fig. S11a). In contrast, EMBEDR identified dubious cell embeddings in both clusters (Supplementary Fig. S11g), but the dubious embeddings and trustworthy embeddings identified by EMBEDR did not exhibit obvious differences in gene expression levels (Supplementary Fig. S11i). Since both scDEED and EMBEDR unified the two clusters in their respective optimized embeddings (Fig. 3b for scDEED; Supplementary Fig. S11h for EMBEDR), scDEED's dubious detection result under the original perplexity 40 was more reasonable than EMBEDR's because the *neuron ec1* cells in each cluster should be either jointly dubious (far away from similar cells) or jointly trustworthy (close to similar cells).

Regarding optimizing the perplexity hyperparameter on the Hydra dataset, since EMBEDR does not have a default list of candidate perplexity values, we provided EMBEDR with the default candidate perplexity values of scDEED, and EMBEDR selected the highest candidate perplexity of 410 (Supplementary Fig. S12). This result is expected because EMBEDR's loss function is the t-SNE loss function (i.e., the KL divergence), and it was reported that the t-SNE loss function tends to decrease as the perplexity increases[24]. That is, EMBEDR is expected to choose the largest candidate perplexity—a conceptually undesirable property.

Although scDEED and EMBEDR found different optimized perplexity values on the Hydra dataset, the resulting t-SNE visualizations were highly similar (Supplementary Fig. S12). Hence, we further compared the two t-SNE visualizations by evaluating two metrics regarding the preservation of neighboring information, as in refs. 20,44. The first metric is the K-nearest neighbors (KNN), which reflects the preservation of local information, i.e., the average proportion of the $K = 10$ nearest neighbors in the pre-embedding space that remain in the set of $K = 10$ nearest neighbors in the 2D-embedding space (a proportion is calculated for every cell, and the average is taken over all cells' proportions). The second metric is the K-nearest clusters (KNC), which reflects the preservation of global information, i.e., the average proportion of the $K = 4$ nearest clusters in the pre-embedding space that remain in the set of $K = 4$ nearest clusters in the 2D-embedding space (a proportion is calculated for every cell, and the average is taken over all cells' proportions). For the Hydra dataset, the embeddings optimized by EMBEDR and the embeddings optimized by scDEED had the same KNC of 0.44, while the KNN was slightly better for scDEED (0.77) than EMBEDR (0.75). Thus, the two sets of optimized embeddings shared similar levels of information preservation.

Despite the similarity of the optimized embeddings produced by scDEED and EMBEDR on the Hydra dataset, scDEED far outperformed EMBEDR in terms of computational efficiency. Running without parallelization, scDEED completed the analysis (dubious embedding detection and perplexity optimization) in 4 h, while EMBEDR finished in 18.5 h using all available processors (the default setting in EMBEDR).

On the Tabula Muris Consortium marrow dataset used in the EMBEDR paper[25], EMBEDR reported an optimal t-SNE perplexity of 1000, and we were able to reproduce EMBEDR's optimized visualization using the processed dataset of 4821 cells (Supplementary Fig. S13b).

In contrast to EMBEDR, scDEED found an optimal t-SNE perplexity of 100 (Supplementary Fig. S13a). This example again highlights EMBEDR's preference for high perplexity values. Note that the scDEED optimal perplexity of 100 falls within the suggested range of perplexity as 1–10% of the number of cells[20,22], while the EMBEDR optimal perplexity of 1000 is far beyond the range.

Also, on the Tabula Muris Consortium marrow dataset, comparison of EMBEDR's and scDEED's optimized visualizations shows a striking difference in the locations of the cluster of early mouse hematopoietic stem cells expressing genes *Kit, Lin*, and *Sca-1* (referred to as *KLS cells*) and the cluster of *granulocytes* (Supplementary Fig. S13a, b). While these two clusters had a large separation in EMBEDR's optimized visualization (Supplementary Fig. S13b), they shared some neighboring cells in scDEED's optimized visualization (Supplementary Fig. S13a). We randomly picked a *KLS cell* close to *granulocytes* and examined the cell's 50 nearest neighboring cells in the 50-dimensional pre-embedding PC space. We found the *KLS cell* close to its 50 nearest neighbors in scDEED's optimized visualization (Supplementary Fig. S13c), but not in EMBEDR's optimized visualization (Supplementary Fig. S13d), suggesting that scDEED's optimized visualization better preserves neighboring information.

Finally, we used the simulator scDesign3[43] to generate 20 simulated scRNA-seq datasets from a model fitted on a real scRNA-seq dataset of mouse small intestinal epithelial cells (Methods). For each simulated dataset, we used scDEED or EMBEDR to optimize the t-SNE perplexity or marginally optimize the UMAP n.neighbors hyperparameter. We fixed the UMAP min.dist hyperparameter at 0.1 to match the EMBEDR algorithm, which only allows optimization of the t-SNE perplexity or the UMAP n.neighbors. Comparing scDEED's optimized hyperparameters with EMBEDR's (Supplementary Fig. S14a–d), we observed that EMBEDR tends to choose the highest perplexity value (consistent with the previous real data results) and the highest n.neighbors value, while scDEED did not exhibit this conceptually undesirable phenomenon. Moreover, scDEED reduced the number of dubious embeddings to zero for both t-SNE and UMAP, while EMBEDR consistently preferred t-SNE to UMAP due to its usage of the t-SNE loss function as the criterion (Supplementary Fig. S14e). Hence, while scDEED can fairly compare 2D-embedding methods, EMBEDR biasedly favors t-SNE by design.

Comparison of the KNC and KNN metrics (Supplementary Fig. S14f) shows that scDEED outperformed EMBEDR in preserving the local information (i.e., scDEED led to significantly higher KNN values than EMBEDR; one-sided paired $t$ test $p$ value = $7.88 \times 10^{-15}$, $n$ = 20 pairs corresponding to the 20 simulated datasets), and scDEED performed similarly to EMBEDR in preserving the global information (i.e., scDEED and EMBEDR had similar KNC values; two-sided paired $t$ test $p$ value = 0.208, $n$ = 20 pairs corresponding to the 20 simulated datasets). These results were consistent with the results of the Hydra dataset presented earlier; the KNC was similar for EMBEDR and scDEED, but the KNN was higher for scDEED than EMBEDR. Hence, scDEED is the better method for preserving neighboring information.

## Discussion

scDEED is a flexible statistical method for detecting cells with dubious embeddings produced by 2D-embedding methods such as t-SNE and UMAP. scDEED detects dubious embeddings based on statistically significant low similarities between ordered neighbors before and after 2D embedding. Based on the detected dubious embeddings, scDEED enables the optimization of an embedding method's hyperparameter setting (e.g., t-SNE's perplexity and UMAP's min.dist and n.neighbors) by minimizing the number of dubious cell embeddings.

Using multiple scRNA-seq datasets and their embeddings from published studies, we demonstrate that the dubious cell embeddings detected by scDEED indeed have dubious locations relative to other cell embeddings. We also show that the hyperparameter settings

optimized by scDEED lead to cell embeddings that better align with biological knowledge and pre-embedding cell distances compared with the original hyperparameter settings used in published studies.

By default, scDEED sets a conservative threshold for detecting dubious cell embeddings: cells are flagged as dubious if their reliability scores are no greater than the 5th percentile of the null reliability scores. However, users of the scDEED R package can increase or decrease the percentile threshold based on their knowledge and preference so that more or fewer cell embeddings will be flagged as dubious.

Minimizing the number of dubious cell embeddings can also help optimize other hyperparameters, such as the random seed and the learning rate, which are known to have impacts on t-SNE and UMAP visualization[17,21]. Further, detecting dubious cell embeddings may help discern the topology of a dataset. For example, if the number of dubious cell embeddings exhibits a complex trend as a hyperparameter value increases, then the cells might have a complex topology, and users should be cautious when interpreting the 2D visualization.

Our results indicate that scDEED outperforms a competing method, EMBEDR, in terms of dubious embedding detection, hyperparameter optimization, and computational time. EMBEDR uses the t-SNE loss function and thus biasedly favors t-SNE over other embedding methods. Even for t-SNE, EMBEDR's optimization is undesirable because it inherently prefers high perplexity values. Another method, DynamicViz, only evaluates cell embeddings from a stability perspective and cannot decide if cell embeddings are dubious based on pre-embedding cell distances. Moreover, both EMBEDR and DynamicViz are computationally expensive because they require many runs of bootstraps followed by embedding. In contrast, scDEED does not have these limitations: scDEED is fair for all embedding methods, effective at detecting dubious embeddings, and computationally efficient.

An interesting observation from our evaluation of scDEED is that t-SNE and UMAP could have more similar visualizations after scDEED's optimization. This finding questions the common belief that t-SNE preserves global information worse than UMAP and suggests that the lack of hyperparameter optimization might be a contributing factor. It is important to note that while we demonstrate the utility of scDEED on scRNA-seq data using t-SNE and UMAP as proof of concept, scDEED is applicable to other data types (e.g., multimodal assays[45]) and other embedding methods as well.

We expect scDEED to be a valuable computational tool for single-cell researchers to generate and interpret visualization plots, which play an essential role in observation-based scientific discoveries. However, open questions remain. First, we lack an understanding of the difference between dubious embeddings and trustworthy embeddings in terms of gene expression profiles. Our empirical evidence suggests that the difference depends on the hyperparameter value(s) at which the dubious embeddings are detected. For example, in the Hydra dataset, at a low t-SNE perplexity, cells with dubious embeddings tend to exhibit higher sparsity compared to cells with trustworthy embeddings. However, this trend reverses after the perplexity surpasses the optimal perplexity selected by scDEED. At the optimal perplexity, cells with trustworthy or dubious embeddings demonstrate statistically similar levels of sparsity (as shown in Supplementary Fig. S15). This result might indicate that the difference between dubious embeddings and trustworthy embeddings is more due to randomness instead of biological signals under the optimized hyperparameter(s). As a future direction, exploring how the difference changes along with the hyperparameter value(s) might offer an alternative way for choosing the hyperparameter value(s). Second, it remains an open question to understand how the optimized hyperparameter setting is related to the algorithms of t-SNE and UMAP. For example, t-SNE uses the perplexity to define cells' neighbor-picking probabilities in the pre-embedding space, and t-SNE's optimization tries to find cells' 2D embeddings so that the cells' neighbor-picking

probabilities are well preserved in the 2D-embedding space. Hence, an open question is whether the optimized perplexity leads to more reasonable neighbor-picking probabilities that better align with cells' topology in some sense.

## Methods

### The scDEED algorithm

Given a gene-by-cell matrix (after appropriate normalization and logarithmic transformation, i.e., the input into PCA with users' discretion) with $n$ columns (corresponding to $n$ cells) and a 2D-embedding method, e.g., t-SNE and UMAP, with a given hyperparameter setting, scDEED finds dubious cell embeddings in the following six steps.

Step 1. scDEED constructs a permuted data matrix by independently permuting the $n$ cells for each gene (i.e., independently shuffling the $n$ values in each row) in the original data matrix.

Step 2 (optional). In most single-cell data analysis pipelines, e.g., the Python package Scanpy[46] and the R package Seurat[47], users perform PCA with a chosen $K$ (the number of PCs) on the original data matrix before applying the 2D-embedding method. Similarly, scDEED asks users to input $K$, and it performs PCA with $K$ on both the original data matrix and the permuted data matrix. Users may choose $K$ based on their preferred method. For the results of this study, we chose $K$ following the original studies or based on the elbow plot if the original studies did not provide $K$.

Step 3. scDEED applies the 2D-embedding method (with a given hyperparameter setting and rand.seed = 100) to the original and permuted pre-embedding matrices. These matrices are either the original data (if Step 2 is not performed) or the $K$ PCs (so each matrix has dimensions $K \times n$). Hence, each cell receives two 2D embeddings, one original and one permuted.

Step 4. Based on the original data before and after the 2D embedding, scDEED defines a *reliability score* for each cell $i = 1, \ldots, n$, based on cell $i$'s $x\%$ (default $x = 50$, the only hyperparameter of scDEED) closest neighbors in the 2D-embedding space and those in the pre-embedding space (the PC space if Step 2 is performed or the original space otherwise), with the neighbors in each space defined based on the Euclidean distance. Given the two sets of neighbors, scDEED calculates cell $i$'s Euclidean distances to the ordered neighbors (from the closest to the farthest) in each set in the 2D-embedding space, obtaining two distance vectors of length $x\% \times n$ (rounded to the closest integer). Finally, scDEED defines the reliability score as the Pearson correlation of the two distance vectors. That is, each cell's reliability score ranges from −1 to 1; a higher reliability score indicates a better agreement between the cell's ordered neighbors before and after the 2D embedding. We use the Pearson correlation because the actual values of the Euclidean distances in the 2D embedding space matter (for our visualization and interpretation), not just the ranks of the Euclidean distances used in the Spearman correlation.

Step 5. Based on the permuted data before and after the 2D embedding, scDEED applies the same procedure in Step 4 to obtain the *null reliability scores* of the $n$ cells. Because of the permutation, the similarities among cells are disrupted, and no biological neighboring relationships are preserved by the 2D embedding. Hence, each cell's neighbors are purely determined by random chance, and its reliability score reflects the random agreement between its ordered neighbors before and after the 2D embedding. Leveraging the $n$ null reliability scores, scDEED finds the thresholds for calling a cell's reliability score low or high.

Step 6. scDEED defines *dubious cell embeddings* as the embeddings of the cells whose reliability scores are less than or equal to the 5-th percentile of the $n$ null reliability scores. On the other end, scDEED defines *trustworthy cell embeddings* as the embeddings of the cells whose reliability scores are greater than or equal to the 95-th percentile of the $n$ null reliability scores.

After the above steps, scDEED reports the number of dubious cell embeddings given a parameter setting. From a grid search of candidate hyperparameter settings, scDEED finds the setting that minimizes the number of dubious cell embeddings.

In the scDEED R package, the following candidate hyperparameter values are set by default, but users can specify their own candidate hyperparameter values. For t-SNE, the default candidate perplexity values are 20, 50, …, 380, 410, 450, 500, …, 750, and 800. For UMAP, the default n.neighbors values are 5, 6, …, 29, 30, 35, 40, 45, and 50; the default min.dist values are 0.0125, 0.05, 0.1, 0.2, …, 0.7, and 0.8.

### Analysis of the effectiveness of the permutation strategy

We illustrate the effects of permutation in removing cell-cell relationships by permuting the *inDrops* dataset twice (with different random seeds). First, we examined the PCA plots of the original *inDrops* dataset and the two permuted datasets (Supplementary Fig. S16a). We observed that the annotated cell types were distinguishable in the original PCA plot, while in the two permuted PCA plots, all cell types were mixed. Next, we examined the t-SNE plots (at the perplexity of 40) of the three datasets and observed a similar loss of cell type patterns in the permuted datasets (Supplementary Fig. S16b). Lastly, we examined the gene expression levels in the three datasets (Supplementary Fig. S16c). We observed that the annotated cell types exhibited clustered patterns of gene expression profiles in the original dataset; however, these patterns disappeared in the permuted datasets. Hence, we conclude that permutation is effective for removing cell-cell relationships in real data.

### Sensitivity analysis of scDEED's only hyperparameter "similarity percent"

Steps 4 and 5 of the scDEED algorithm require the only hyperparameter of scDEED, $x$, the "similarity percent" (i.e., the percentage of closest neighbors, or the neighborhood size). The default value is $x = 50$, meaning that half of all cells are considered as neighbors. Intuitively, a smaller value of $x$ defines a smaller neighborhood size and would thus place a greater emphasis on preserving local structures. To investigate the effect of $x$ on the performance of scDEED, we examined $x = 5, 20, 35, 50, 65, 80$, and 95 on the Hydra dataset with t-SNE as the embedding method.

First, we examined the numbers of dubious and trustworthy cell embeddings found at each $x$ when applying scDEED under a range of t-SNE perplexity values (Supplementary Fig. S17a–c). At a small similarity percent $x = 5$, the number of dubious cell embeddings was relatively stable across the perplexity values, an expected result as t-SNE was designed to preserve cells' local neighborhoods. At a large similarity percent $x = 95$, the number of trustworthy embeddings was smaller than at the other $x$ values under most perplexity values; this result was also expected because t-SNE was not designed to preserve cells' global topology. Hence, using a too small or large $x$ would not reflect t-SNE's ability to preserve mid-range neighbors or adjacent cell clusters' relative positions. This result justified the default value of $x = 50$ in scDEED.

Second, we observed that, at any $x$, as the perplexity value increased past a threshold (around 170 for the Hydra dataset), the number of dubious cell embeddings tended to stay stable and did not decrease further (Supplementary Fig. S17a). This result provided evidence that scDEED does not have a bias towards large perplexity values. Notably, at the original perplexity, too small or large $x$ values found fewer dubious cell embeddings than the $x$ values around the default $x = 50$ did, again implying that too small or large $x$ values are unsuitable for detecting dubious cell embeddings (Supplementary Fig. S17d). Most importantly, the $x$ values around 50 resulted in optimized perplexity values that were similar, confirming the stability of scDEED with the default $x = 50$ (Supplementary Fig. S17e).

To further explain the above observations, we examined the distributions of reliability scores in the permuted data and the original

data at each $x$. We found the distributions of null reliability scores (i.e., reliability scores of permuted cells) to be stable across $x$ values despite exhibiting a slight monotone shift to the right as $x$ increased (Supplementary Fig. S17f). On the original data, the distribution of reliability scores of unpermuted cells was most concentrated on lower scores when $x = 95$ (Supplementary Fig. S17g). This explains why $x = 95$ found the fewest cell embeddings to be trustworthy.

Third, we examined the dubious or trustworthy cell embeddings detected by scDEED at $x = 5$, 50, or 95 from the original Hydra embeddings. Several clusters in the original embeddings were detected as dubious at both $x = 50$ and 95, but not at $x = 5$ (Supplementary Fig. S18a). Also, only at $x = 5$, almost all cell embeddings were found as trustworthy (Supplementary Fig. S18b). We believe that the different detection result at $x = 5$ was due to the fact that examining too small neighborhoods was ineffective in revealing the clusters that had dubious positions relative to other clusters.

Fourth, we further examined the similarities of dubious cell embeddings detected at different $x$ values given a perplexity value. We focused on dubious cell embeddings because their number is the criterion scDEED uses for optimization. We considered three perplexity values covering a wide range: perplexity 40 (used for the original embeddings), perplexity 230 (optimized by scDEED at $x = 50$), and perplexity 410 (the maximum candidate perplexity value in the scDEED package) (Supplementary Fig. S19). Given each perplexity value, we used scDEED to detect a set of dubious cell embeddings at each $x$ value; then we calculated the Jaccard index between the two sets for every pair of $x$ values (Supplementary Fig. S19). We observed that the dubious embeddings detected at too small $x$ values had little-to-no agreement with the dubious embeddings detected at other $x$ values, an undesirable result as we would expect the dubious embeddings to be reasonably robust to the $x$ value. In contrast, middle-to-high $x$ values ($x = 50$, 65, and 80) tended to have high agreement with each other, particularly $x = 50$. We also examined three $x$ values close to 50 ($x = 40$, 50, 60) and confirmed that their respectively optimized visualizations were highly similar (Supplementary Fig. S20).

In conclusion, the above sensitivity analysis results supported our default choice of $x = 50$. A similar rationale is described in ref. 27, which found that effective dimension reduction required emphasis on mid-range neighbors.

### Alternative hyperparameter optimization via "kneedle"

Instead of looking for the hyperparameter value (e.g., the t-SNE perplexity) to minimize the number of dubious embeddings over a default grid of candidate hyperparameter values, we implemented the "kneedle" method that searches for the hyperparameter value as the elbow point in the plot of the number of dubious embeddings (i.e., the y-axis) versus the hyperparameter value (i.e., the x-axis)[30]. We investigated this alternative optimization approach on two datasets and found that the resulting t-SNE visualizations were highly similar to those resulted from the grid-search global min approach used in scDEED (Supplementary Figs. S3–4, 20).

We also compared the two hyperparameter optimization approaches (the global min approach used in scDEED and the "kneedle" method) for three $x$ values (40, 50, and 60) on the Hydra dataset (Supplementary Fig. S20). We found that for each $x$ value, the two approaches' optimized perplexity values led to highly similar t-SNE visualizations. The results confirm that scDEED's optimized visualization based on the number of dubious embeddings is not sensitive to the optimization approach.

### Implementation of t-SNE and UMAP

We performed t-SNE and UMAP using the functions `RunTSNE()` and `RunUMAP()` respectively in the R package Seurat (version 3.2.3). The hyperparameters scDEED optimizes are `perplexity` in the `RunTSNE()` function and `n.neighbors` and `min.dist` in the `RunUMAP()` function. We used "seed.use = 100" when running `RunTSNE()` and `RunUMAP()` and kept the rest of the arguments as default.

### Assessing the purity of cell clusters in the Hydra dataset

We used the function `CalculateRogue()` in the R package Rogue (version 2.0.0) to calculate the ROGUE statistic, which measures the purity of a cell cluster. The larger the ROGUE value, the purer (or more homogeneous) the cell cluster. In the Hydra dataset, the ROGUE values of the five clusters of neuron ectodermal cells are 0.710 (*neuron ec1*), 0.784 (*neuron ec2*), 0.714 (*neuron ec3*), 0.793 (*neuron ec4*), and 0.839 (*neuron ec5*).

### Calculation of ARI

We used the function `ARI()` in the R package aricode (version 1.0.2) to calculate the adjusted Rand index (ARI)[39], which represents the agreement between two sets of labels (e.g., a set of cells' cluster labels and a set of cells' annotated type labels) with adjustment for chance agreement of labels. ARI ranges from 0 to 1, with 1 indicating perfect agreement and 0 indicating no agreement beyond random chance.

### Selection of genes in heatmaps

In every heatmap, unless otherwise specified, we plotted the top 300 genes that have the largest expression variances (based on the gene expression values before the PCA step in the Seurat package) across the cells shown in the heatmap.

### DE gene identification

Differentially expressed (DE) genes were found using `FindMarkers()` function in the R package Seurat (version 4.3.0.1) with the Wilcoxon rank-sum test (the default setting).

### Evaluation metrics for local and global preservation

We evaluated the preservation of information as in ref. 20, using the following two metrics.

KNN reflects the preservation of local information, i.e., the average proportion of the $K = 10$ nearest neighbors in the pre-embedding PC space that remain in the set of $K = 10$ nearest neighbors in the 2D-embedding space. KNN was also used in ref. 44.

K-nearest clusters (KNC) reflects the preservation of global information, i.e., the average proportion of the $K = 4$ nearest clusters in the pre-embedding PC space that remain in the set of $K = 4$ nearest clusters in the 2D-embedding space. Note that for KNC, we deviated from ref. 20 by defining each cluster center as the median rather than the mean because the median is more robust to outliers.

### Datasets

Whenever preprocessed datasets were available, they were directly used in this study. Otherwise, datasets were preprocessed in the same way as in the original studies that generated the data. Below is the preprocessing detail for every dataset. The preprocessed data and code are available on Zenodo (https://zenodo.org/record/7216361#. ZDNgd-zMLJ8).

### Hydra

The dataset Hydra/Hydra_Seurat_Whole_Transcriptome.rds (from the original study) contains the transcriptomes of $n = 25{,}052$ single Hydra polyp cells sequenced by Drop-seq, with the cells labeled with cluster labels, and 33,391 genes' scaled expression levels processed by Seurat[28]. The data from the original study was archived at NCBI GEO with the accession code GSE121617. Following the original study, we used $K = 31$ PCs in Step 2 of scDEED, and we used the default `RunTSNE(dims = 1:5)` in the Seurat R package as the 2D-embedding method. The preprocessing code is in Hydra/data_processing_hydra.Rmd, and the preprocessed dataset is in Hydra/Hydra.rds.

We ran scDEED with the candidate perplexity values 10, 30, …, 390, and 410, as well as the value 40 used in the original study. The running time was 2.80 hours (see "Computing environment").

## CAR-T

In patients with B cell malignancies, lymphodepletion chemotherapy followed by infusion of CD19-specific chimeric antigen receptor modified-T (CAR-T) cells is known to generate anti-tumor responses. The dataset was produced to understand the clonal composition of CAR-T cells in the infusion products (IP) after the adoptive transfer. In particular, the dataset contains a sample of 10 patients who received CD19-specific CAR-T cells, and it is representative of the population in terms of age, sex, adverse events, clinical outcome, lymphodepletion therapy, and cell dose. Using the 10x Genomics platform, single-cell RNA-seq data were generated from $n = 62,167$ CD8 + CAR-T cells sorted based on truncated human epidermal growth factor receptor (EGFRt) expression from the IP and blood at the early (day 7–14), late (day 26–30), and very late (day 83–112) time points after infusion. This dataset is in the file CART/raw.expMatrix.csv, downloaded from the Gene Expression Omnibus (GEO) with the accession code GSE125881. Following the original study, we used $K = 15$ PCs in Step 2 of scDEED, and we used the default `RunTSNE(dims = 1:5)` in the Seurat R package as the 2D-embedding method. The preprocessing code is in CART/data_processing_CART.Rmd, and the preprocessed dataset is in CART/seuratObj_v3.RData.

We ran scDEED with the candidate perplexity values 20, 50, …, 380, 410, 450, 500, …, 750, and 800, as well as the value 30 used in the original study. The running time was 14.37 h (see "Computing environment")

## Alveolar

This dataset was constructed to learn the cell-cell communication during the regeneration process after bleomycin-induced lung injury. It contains whole-organ single cell suspensions from mice, from six-time points after injury and uninjured control lungs with four replicate mice per time point on average. Single-cell transcriptomes from about 1000 cells per individual mouse were carried out using the Dropseq workflow, leading to a sample of $n = 29,297$ cells in the final dataset. The raw dataset is available at NCBI GEO with the accession code GSE141259. Following the original study, we used $K = 50$ independent components in Step 2 of scDEED to obtain the pre-embedding space prior to applying UMAP in the Seurat R package, `RunUMAP(dims = 1:50)`. Unlike `RunTSNE`, Seurat requires the user to specify the input dimension for UMAP. The preprocessing code is in Alveolar/data_processing_Alveolar.Rmd, and the preprocessed dataset is in Alveolar/Seurat_v3.RData.

We ran scDEED with the candidate n.neighbors values 5, 6, …, 9, 10, 15, 20, …, 45, 50, 80, 160, …, 240, and 320; and candidate min.dist values 0.0125, 0.05, 0.1, 0.2, …, 0.7, and 0.8. The running time was 1.37 h for marginal optimization of n.neighbors, 38.59 min for marginal optimization of min.dist, and 12.42 h for joint optimization (see "Computing environment").

## Samusik

Cells were gathered from bone marrow samples, and cell surface markers were used for CyTOF analysis. Data were normalized and annotated with clusters and the hand-gated populations. Doublets and neutrophils were removed. The original dataset is available at [https://figshare.com/s/9c3a0136f12b97f1dadd][14]. The final dataset has $n = 841,644$ cells. Following the original study, we used $p = 38$ genetic markers as the pre-embedding space (optional Step 2 was omitted) before applying UMAP using the R command `RunUMAP(features = feature_list)`, where feature_list refers to the 38 markers. The preprocessing code is in Samusik/data_processing_samusik01.Rmd, and the preprocessed dataset is in Samusik/samusik01_seurat.Rdata.

We ran scDEED with the same candidate n.neighbors and min.dist values as for the Alveolar dataset. The running time was 12.43 h for marginal optimization of n.neighbors, 4.89 h for marginal optimization of min.dist, and 3.77 days for joint optimization (see "Computing environment").

## Human PBMC

This dataset was gathered by the Broad Institute to compare seven single cell/single nucleus sequencing methods[38]. The original study manually annotated cells based on canonical cell markers. Here, we focused on three sequencing methods (inDrops, DropSeq, and Seq-Well) and four common cell types *Cytotoxic T cell, CD4 + T cell, CD14+ Monocyte*, and *B cell*. This resulted in $n = 5858$ cells for inDrops, $n = 5801$ cells for DropSeq, and $n = 3626$ cells for SeqWell. The entire dataset is available as pbmcsca.SeuratData in the R package SeuratData (also available at NCBI GEO with the accession number GSE132044). The subset of data we analyzed is in Across_Techniques/Seurat.Rdata. We used $K = 50$ PCs in Step 2 of scDEED. The 2D-embedding space was obtained using `RunUMAP(dims = 1:50)` and `RunTSNE(dims = 1:5)`.

We ran scDEED with the candidate perplexity values 5, 10, …., 135, and 140; n.neighbors values 5, 6, 7, 8, 9, 10, 20, 30, 40, 50, 80, 160, and 240; and min.dist values 0.0125, 0.05, 0.1, 0.3, 0.5, 0.7, and 0.9.

## Marrow

This dataset was used in the EMBEDR paper[25] and is a subset of a single-cell transcriptome of *Mus. musculus*[42]. Cells were harvested from mice and sorted with fluorescent-activated cell sorting (FACS). Sequencing was done using the Smart-seq2 protocol with Illumina sequencing. The original dataset is available as Marrow/Marrow_counts at

[https://figshare.com/projects/Tabula_Muris_Transcriptomic_characterization_of_20_organs_and_tissues_from_Mus_musculus_at_single_cell_resolution/27733]. The original dataset contained $n = 5037$ cells. Following the preprocessing notebook available at EMBEDR's GitHub (Marrow/Marrow preprocessing.ipynb), we obtained $n = 4821$ cells (Marrow/Marrow_processed.csv). This differs from EMBEDR's reported $n = 4771$ cells after the preprocessing (in the EMBEDR publication). However, the EMBEDR's authors' code indicated that all $n = 5037$ cells were used for analysis.

Despite this discrepancy, our preprocessed $n = 4821$ cells with 17,303 genes replicated the EMBEDR results fairly well. For fair comparison, the processed data was used for all analyses. Following the EMBEDR tutorial at

[https://github.com/ejohnson643/EMBEDR/blob/master/projects/Figures/Figure_04v1_GlobalParameterSweep.ipynb], we used $K = 50$ PCs as the pre-embedding space prior to EMBEDR optimization. For scDEED optimization, we used $K = 16$ PCs (chosen from an elbow plot) in Step 2 of scDEED. The low dimensional space and visualizations were obtained using the default Seurat R command `RunTSNE(dims = 1:5)`.

We ran scDEED with the candidate values 10, 15, 20, 30, 40, 50, 60, 80, 100, 150, 200, 300, 350, 500, 600, 800, 1000, 1300, 1700, 2200, 2900, and 3700 to mimic the $K_{eff}$ values in the original Fig. 4 in the EMBEDR paper[25]. The running time was 10 min (see "Computing environment").

## DG

This dentate gyrus dataset was measured to elucidate the gyrus cell lineage. The 10x Genomics processed data used in the tutorial of the R package Velocyto (version 0.6) was analyzed (Velocyto/10×43.1_loom). The tutorial is available at

[https://htmlpreview.github.io/?https://github.com/satijalab/seurat.wrappers/blob/master/docs/velocity.html].

Cells were annotated using the Louvain clustering at the default resolution (0.8) based on the marker genes from the original paper

(Velocyto/data_annotated.Rds). The dataset consists of $n = 3396$ cells and 92,135 features across the spliced and unspliced assays and is accessible at
[http://pklab.med.harvard.edu/velocyto/DG1/10X43_1.loom]. For scDEED optimization, we used $K = 12$ PCs in Step 2 of scDEED and obtained the low dimensional space using the default Seurat R command `RunTSNE(dims = 1:5)`. Final visualization used the command `RunTSNE(dims = 1:12)`.

We ran scDEED with the candidate perplexity values 20, 50, ..., 380, 410, 450, 500, 600, 700, and 800.

### Simulated data
The 20 simulated datasets (Simulated_Data/Simulated_data_1.Rds, ..., Simulated_Data/Simulated_data_20.Rds) were generated by scDesign3[43], which was trained on a built-in dataset of mouse small intestinal epithelial cells of the R package scDesign2[48] (GEO accession code GSE92332[49]). To increase the distances between three cell types (*Enterocyte. Progenitor*, *TA.Early*, and *Stem*), we independently permuted each cell type's gene expression mean values (every gene has a mean parameter value in each cell type) across all genes. To ensure that gene expression did not largely deviate from the specified cluster means, the 100 largest dispersion parameters were divided by 150. In total, we had 10,000 genes and $n = 7217$ cells. For scDEED optimization, we used $K = 12$ PCs in Step 2 of scDEED. To obtain the 2D-embedding space, we used `RunUMAP(dims = 1:12)` and the default `RunTSNE(dims = 1:5)`. For EMBEDR, we also used $K = 12$ PCs for its optimization.

To compare t-SNE and UMAP fairly, we calculated the final 2D embeddings, which were used for KNN and KNC comparison, all using $K = 12$ PCs as the pre-embedding space.

We ran scDEED with the candidate perplexity values 20, 50, ..., 380, 410, 450, 500, ..., 750, and 800 (default settings in scDEED).

### RNA velocity
RNA velocity was performed using Velocyto (version 0.6) with default settings using the tutorial available at:
[https://htmlpreview.github.io/?https://github.com/satijalab/seurat.wrappers/blob/master/docs/velocity.html].

### Comparison with EMBEDR and DynamicViz
EMBEDR is available as a Python package. Hyperparameter sweeps were performed following the available tutorials with default settings, including the option to use all available processors. Applying EMBEDR to the Hydra dataset, we used the suggested 25 data embeddings with 15 null embeddings. Applying EMBEDR to the 20 simulated datasets, we used 5 data embeddings with ten null embeddings to save computational time. Since EMBEDR requires the user to provide a list of candidate hyperparameter parameters, we used the default lists of perplexity and n.neighbors values in scDEED. EMBEDR does not sweep over min.dist, so for a fair comparison, we fixed min.dist at 0.1 (default EMBEDR setting) when using scDEED.

EMBEDR categorizes cells as well-embedded or noisy. For consistency in terminology between EMBEDR and scDEED, we considered well-embedded cells to have trustworthy embeddings and noisy cells to have dubious embeddings. Specifically, we defined dubious cell embeddings to be the cells with EMBEDR $p$ values above 0.1 based on the EMBEDR paper[25], which considers all cells with $p$ values > 0.1 to have similar levels of noise.

DynamicViz is also available as a Python package. Parameter sweeps are not built in but can be iterated. We were able to successfully use this package for t-SNE, yet for UMAP there were some errors. Due to this and the conceptual difference in the definition of dubious cell embeddings between DynamicViz and scDEED, we decided to omit DynamicViz from our analysis.

### Statistics and reproducibility
No statistical method was used to predetermine the sample size. For reproducibility, please refer to the Zenodo deposit ([https://zenodo.org/record/7216361#.ZDNgd-zMLJ8]) for all the code used to generate figures, as well as the processed datasets.

### Versions of R packages
**Seurat** version 4.3.0.1: all the t-SNE and UMAP analyses except the EMBEDR analysis.
**SeuratData** version 0.2.2: for the dataset "pmbcsca.SeuratData".
**doParallel** version 1.0.15; **foreach** version 1.5.0: for parallel computing and looping.
**ggsci** version 2.9: for plotting.
**Rogue** version 2.0.0: for assessing the purity of a cell cluster.
**distances** version 0.1.8: fast computation for pairwise distances between vectors.
**velocyto.R** version 0.6: RNA velocity analysis.
**scDesign3** version 0.99.6 for data simulation.
Other packages:
**Rfast** version 1.9.9; **VGAM** version 1.1.3; **pracma** version 2.2.9; **ggplot2** version 3.3.2; **SeuratWrappers** version 0.3.0; **aricode** version 1.0.2

### Computing environment
All algorithms and code were executed on an iMac with 3.6 GHz Intel Core i9 processor, 64GB memory, and Mojave 10.14 system. For the data analysis performed in this paper, six cores were used.

### Reporting summary
Further information on research design is available in the Nature Portfolio Reporting Summary linked to this article.

## Data availability
All relevant data supporting the key findings of this study are available within the article and its Supplementary Information files. All processed datasets are available at [https://zenodo.org/record/7216361#.ZDNgd-zMLJ8]. The original datasets are listed below. The Hydra dataset was obtained with the NCBI GEO accession code GSE121617. The CAR-T dataset was obtained with the NCBI GEO accession code GSE125881. The Alveolar dataset was obtained with the NCBI GEO accession code GSE141259. The Samusik dataset was obtained from [https://figshare.com/s/9c3a0136f12b97f1dadd]. The Human PBMC dataset was obtained from the SeuratData package, version 0.2.2, and is also available with the NCBI GEO accession code GSE132044. The Marrow dataset was obtained from [https://figshare.com/projects/Tabula_Muris_Transcriptomic_characterization_of_20_organs_and_tissues_from_Mus_musculus_at_single_cell_resolution/27733]. The dentate gyrus dataset was obtained from [http://pklab.med.harvard.edu/velocyto/DG1/10X43_1.loom]. Data (mouse small intestinal epithelial cells) with the NCBI GEO accession code GSE92332, was used to train scDesign2 to simulate data. The reproducibility material (including datasets, R and Python scripts, and intermediate results) is provided with this paper through Zenodo, [https://zenodo.org/record/7216361#.ZDNgd-zMLJ8]. Source Data files (Excel and CSVs sufficient to generate dot, line, and box plots) are provided with this paper through Zenodo, [https://zenodo.org/records/10511446].

## Code availability
The scDEED R package is available at the GitHub repository [https://github.com/JSB-UCLA/scDEED] (Zenodo doi: [https://zenodo.org/badge/latestdoi/402656304])[50] The computer code is available at Zenodo [https://zenodo.org/record/7216361#.ZDNgd-zMLJ8].

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

## Acknowledgements

This work was supported by the following grants: Hong Kong RGC Grant project number 26305120 (to L.X.); NSF grant DGE-1829071 (to C.L.); National Science Foundation DBI-1846216 and DMS-2113754, NIH/NIGMS R01GM120507 and R35GM140888, Johnson & Johnson WiSTEM2D Award, Sloan Research Fellowship, UCLA David Geffen School of Medicine W. M. Keck Foundation Junior Faculty Award, and Chan-Zuckerberg Initiative Single-Cell Biology Data Insights [Silicon Valley Community Foundation Grant Number: 2022-249355] (to J.J.L.). The authors would like to thank Dr. Xin Tong at University of Southern California and Dr. Yanhui Wu at Hong Kong University for an initial discussion of the project idea. The authors would also like to thank Mr. Tianyang Liu at UCLA for assisting with the scDEED R package development and Mr. Dongyuan Song at UCLA for suggesting relevant literature.

## Author contributions

L.X. and C.L. are co-first authors. J.J.L. and L.X. conceived of the study. L.X. and C.L. performed data analysis. L.X. and C.L. developed the scDEED R package. L.X., C.L., and J.J.L. wrote the manuscript.

## Competing interests

The authors declare no competing interests.
