## [Peer Review File · Nature Communications]

Statistical method scDEED for detecting dubious 2D single-cell embeddings and optimizing t-SNE and UMAP hyperparametersReviewer #1 (Remarks to the Author):

The manuscript reported a refining method for the commonly used 2D embedding methods for single-cell data visualization, with the purpose to identify dubious cell embeddings from the results of t-SNE, UMAP or other cell 2D embedding methods. The idea is valuable, however, the authors failed to provide convincing evidence to prove that the scDEED method as developed by the authors could reliably identify dubious cell embeddings.

Major questions:

1. The basic idea of scDEED is to comparing the reliability of cell neighbors after 2D embedding using randomly permuted cell matrix as a reference, using Euclidean distance to define cell neighbors. There are two concerns related to this design: 1) Due to the large size of single cell data, random permutation could still generate some cell relationships similar or identical to the real case; 2) Euclidean distance is not excel in representing high dimensional relationships, using Euclidean distance to evaluate 2D embedding outcomes could still generate unreliable results.
2. The authors only used gene expression heatmaps to demonstrate that scDEED could generate more reliable 2D embedding results, which are not convincing. In addition, the heatmap results shown by the authors are also questionable. For example, both the dubious cell and trustworthy cells shown in Supplementary Figure 1a were scattered in different cell clusters, how could the dubious cells or trustworthy cells from different cell clusters have highly similar gene expression profile in Supplementary Figure 1b?
3. The authors only chose cells with > 95th percentiles of null reliability scores as trustworthy embeddings, and treat cells with < 5th percentile of the null reliability scores as dubious embeddings, then how do they deal with the vast majority of cells with null reliability scores between 5th ~ 95th percentiles? Why in most figures, the proportion of trustworthy embeddings is much larger than 95th percentiles?
4. For the same dataset shown in Figure 2c and Supplementary Figure 1a, why did the number of trustworthy cells differ so greatly with the same parameter setting?

Reviewer #2 (Remarks to the Author):

t-SNE and UMAP are arguably the most widely used data visualization methods in single-cell genomics data analysis. While they are effective in manifesting clustered patterns of high-dimensional single-cell genomics data, such embedding methods are notoriously confusing for biomedical researchers to interpret due to the non-linear nature of the t-SNE and UMAP dimensionality reduction algorithms, resulting in unpreserved distances between data points after embedding. A computational approach to quantify the level of reliability or trustworthiness of such embedding methods is crucial and needed in the field.

In this manuscript, Xia et al. present a statistical method for identifying dubious embeddings for single cells in 2D visualizations such as t-SNE and UMAP. This method calculates a correlation score for each data point in an embedding dataset to quantify the embedding's reliability and identifies dubious data points with untrustworthy embeddings based on a statistical model with permutation-generated null distributions. This method can be used to optimize hyperparameter settings in embedding methods. The method is well designed with simple and clear ideas as well as rigorous statistical assessments. The authors have clearly shown the effectiveness of this method in several real scRNA-seq datasets in detecting unreliably placed cells or improving embedding analysis. The manuscript is clearly written, and the data presented are convincing. Based on the data presented in the manuscript, I believe every researcher who analyzes single-cell genomics data should use scDEED to evaluate and optimize t-SNE or UMAP embedding for reliable results before presenting the 2D visualization data. I have no doubt that this timely method should and will be widely used in the field.

I have some minor comments that the authors may want to consider for improving the clarity of the manuscript.

1. For Figure 1, it might be beneficial to include additional graphs to provide a more detailed representation of the scDEED procedure. For example, scatter plots showing the original and

embedding distances (in x and y) of a cell to its $n/2$ neighbors for Pearson correlation calculation could be added for cell 1 and cell 2 to demonstrate the different levels of correlation between the two example cells. Additionally, plotting the reliability score distributions for both real data and permutation data would also be helpful to show the procedure.

2. In Figure 1, the legend currently showing "1st neighbor" and "[$n/2$]th neighbor" is potentially confusing and misleading. Instead, it would be clearer to show a full color palette indicating the continuous color scale from the 1st neighbor to the $n/2$ th neighbor. As there are more than 2 colors involved in representing the full spectrum of the $n/2$ neighbors, a color palette would be more appropriate.

3. In Figures 2, 6, & 8, Panels a & d in each figure, the cell type labels on the t-SNE or UMAP plots are difficult to read. It is recommended that the cell types be labeled on a separate color legend.

4. In Figures 3c, 3f, 3i, 5c, 5f, 5i, 7c, and 7f, the color scales of the heatmaps can be different for different datasets (while being the same for the same dataset/figure), to increase the contrast of the heatmaps and improve visualization.

5. The authors performed a sensitivity analysis of scDEED to its only hyperparameter, x , the percentage of closest neighbors. However, the analysis only explored x values for 40, 50, and 60. It may be interesting for the authors to comment on the possibility of using much smaller or larger x . As the actual distances can be very large across cells from different cell types/clusters, and the Pearson correlation coefficient is sensitive to outliers, if a user only cares about preserving distances within close neighbors (e.g., cells in the same cluster/cell type), would using a very small x (e.g., 5, 10, 20) still be appropriate for calculating the reliability score? Any comments or recommendations from the authors would be helpful.

Reviewer #3 (Remarks to the Author):

This manuscript introduces a permutation-based statistical method/procedure for assessing the reliability of scRNAseq data embedding by reporting the percentage of cells in the dataset with a reliable embedding. The reliability of each cell embedding is evaluated by how well the neighbor distances in the original data are preserved in the embedding compared with randomized cases. This work investigate an important problem in scRNAseq data interpretation and visualization. The current analysis still needs some more investigations to reveal some statistical properties of the permutation procedure. The evaluation and the presentation can also be improved.

1. Lack of understanding of the potential inherent biases in the statistical measures.

1.1 Based on the results, scDEED tends to select larger tSNE smoothness parameter/Gaussian width/perplexity compared with commonly used choices (<50). Given large Gaussian widths often lead to over-smoothing/underfitting of the distance measures, it is very important to investigate the potential inherent biases in the application of the permutation. The plots in Figure S3 and S4 show that the number of dubious cell embeddings is much more in smaller perplexity (<100) than larger ones. Does this suggest a general trend that larger perplexity leads to smaller number of dubious cells and why?

1.2 One more potential bias to the definition of "dubious embedding" is the size of the cell clusters, e.g. in Figure 2b-2c, the dubious cells are almost all in the small clusters (such as "neuron en2"). It appears that the choose of $x\%$ of neighbors used for calculating the neighbor preservation is somewhat undecided. The plots in Figure S15 shows almost no difference with $x\% = 40\%$, 50% and 60% . What happens for smaller $x\%$ and larger $x\%$. Intuitively, $x\%$ should be related to the number and the size of the clusters?

1.3 Another potential bias is the density of the gene expressions in different cell types. Would the density also affect the significance in the permutation? This can be problematic since scDEED might tend to pick cells with lower/higher density as more reliable?

1.4 It will also be interesting to look into how the selection of the perplexity parameter changes the t-distributions of tSNE embedding. Does the parameter tuning leads to a better fitting t-distribution?

2. More evaluation will be helpful.

2.1. Current evaluation are mainly based interpreting the clustering results in each dataset. The KNN and KNC measures were only measured in the simulation data. Based on the ground-truth/human curated labels on the real datasets, does scDEED performs differently from EMBEDR?

2.2 Does the improved 2D embedding in visualization also quantitatively improve the clustering measured by ARI? It will be interesting check it as well.

3. The presentation needs some improvement.

3.1 Some important results for understanding the methods are discussed in the method section instead of the result section. It will be helpful to add a new section on the results of the statistical evaluation of the method.

3.2 The comparison with EMBEDR and the simulation should be presented first such that a reader can understand the result on each individual dataset better given this is a mainly a methodology paper?

Responses to Reviewers' Comments on "scDEED: a statistical method for detecting dubious 2D single-cell embeddings and optimizing t-SNE and UMAP hyperparameters"

We thank the three reviewers for their time reviewing our paper and their insightful, constructive comments. We have thoroughly revised our manuscript based on their comments by improving the clarity of our previous manuscript and including new real data analysis results to support our method.

In summary, we have made the following 4 major changes in our revised manuscript, summarized in Figures 1 and 10, as well as Supplementary Figures S15–S19.

- I. We have improved the clarity of the scDEED algorithm by providing more details in the algorithm illustration as in Figure 1 (suggested by Reviewer 2) and adding a new section at the beginning of Results describing the scDEED algorithm (suggested by Reviewer 3). We have added additional analysis and a new Supplementary Figure 15 to demonstrate that permutation can remove all cell-cell relationships (suggested by Reviewer 1).
- II. We have included extensive analysis on the "similarity percent" (x), scDEED's only hyperparameter (default value is $x = 50$, meaning that the neighborhood size is 50% of all cells), as suggested by Reviewer 2 and Reviewer 3. We have shown that results obtained from too small or large values of x are not reliable, and we have provided more justifications for the default value $x = 50$.
- III. When we compared the performances of scDEED with EMBEDR, we have included more metrics, including KNN (K-nearest neighbors) and KNC (K-nearest clusters) for the *Hydra* dataset, and ARI (adjusted Rand index) for the *Human PBMC* dataset, as suggested by Reviewer 3. These results further supported our previous findings, specifically that EMBEDR and scDEED share similar KNC metrics, but scDEED has better KNN than EMBEDR. Additionally, for the *Human PBMC* dataset, ARI indicates that the scDEED-optimized embeddings better separate cell types than the original embeddings.
- IV. We have made multiple improvements to the figures as suggested by Reviewer 2, including improved figure legends and contrast in heatmaps.

On the following pages, please find our point-by-point responses to the three reviewers' comments. The reviewers' comments are in *italic*. Quotes to the revised manuscript are highlighted in blue.

Reviewer #1 (Remarks to the Author):

The manuscript reported a refining method for the commonly used 2D embedding methods for single-cell data visualization, with the purpose to identify dubious cell embeddings from the results of t-SNE, UMAP or other cell 2D embedding methods. The idea is valuable, however, the authors failed to provide convincing evidence to proof that the scDEED method as developed by the authors could reliably identify dubious cell embeddings.

Response:

We thank the reviewer for the comments, which have led to a significant improvement of the clarity of our manuscript. In the following, we have added results to address the reviewer's concerns, and we have added clarifications to avoid any potential misunderstandings.

Major questions:

1. The basic idea of scDEED is to comparing the reliability of cell neighbors after 2D embedding using randomly permuted cell matrix as a reference, using Euclidean distance to define cell neighbors. There are two concerns related to this design:

- 1. Due to the large size of single cell data, random permutation could still generate some cell relationships similar or identical to the real case;*
- 2. Euclidean distance is not excel in representing high dimensional relationships, using Euclidean distance to evaluate 2D embedding outcomes could still generate unreliable results.*

Response:

To address the reviewer's first concern, we have added numerical results to demonstrate that random permutation (in which every gene has its expression levels permuted across all cells, and different genes' permutations are conducted independently) does remove cell-cell relationships in the real data. That is, after permutation, every cell's neighboring cells become randomly assigned and carry no biological meanings. Specifically, we conducted random permutation on cells in the *inDrops* data (in the *Human PBMC* dataset) and compared the PCA (and t-SNE) cell embeddings before and after the permutation. We observed an apparent loss of cell cluster patterns in the permuted data. We also examined the gene expression matrix and found that all patterns in the matrix were disrupted after the random permutation.

We have added the following **Methods section**.

Analysis of the effectiveness of the permutation strategy

We illustrate the effects of permutation in removing cell-cell relationships by permuting the *inDrops* dataset twice (with different random seeds). First, we examined the PCA plots of the original *inDrops* dataset and the two permuted datasets (Supplementary Figure S15a). We

observed that the annotated cell types were distinguishable in the original PCA plot, while in the two permuted PCA plots, all cell types were mixed. Next, we examined the t-SNE plots (at the perplexity of 40) of the three datasets and observed a similar loss of cell type patterns in the permuted datasets (Supplementary Figure S15b). Lastly, we examined the gene expression levels in the three datasets (Supplementary Figure S15c). We observed that the annotated cell types exhibited clustered patterns of gene expression profiles in the original dataset; however, these patterns disappeared in the permuted datasets. Hence, we conclude that permutation is effective for removing cell-cell relationships in real data.

Supplementary Figure S15 is pasted on the next page for the reviewer's reference.

Regarding the second point raised by the reviewer, we agree that Euclidean distance may not faithfully represent high-dimensional cell-cell relationships. As a clarification, scDEED uses Euclidean distances only in the pre-embedding and post-embedding spaces, both of which are not high-dimensional. The post-embedding space is clearly low-dimensional. The pre-embedding space is typically not the original feature space but a k -dimensional PC space, where k is user-specified and typically between 20 and 50, often selected by the elbow plot or similar methods. In the case where the original dimension is not too large, as in the *Samusik* dataset where only 38 surface proteins were measured, it may be used as the pre-embedding space. In summary, scDEED does not define cell-cell relationships based on Euclidean distances in the original high-dimensional space; instead, scDEED only uses Euclidean distance in the reduced-dimension pre-embedding space (before t-SNE or UMAP) and the 2D post-embedding space. Below is the **paragraph** in the original manuscript where we discussed the use of Euclidean distance.

Step 4. Based on the original data before and after the 2D embedding, scDEED defines a *reliability score* for each cell $i = 1, \dots, n$, based on cell i 's $x\%$ (default $x = 50$, the only hyperparameter of scDEED) closest neighbors in the 2D embedding space and those in the pre-embedding space (the PC space if Step 2 is performed or the original space otherwise), with the neighbors in each space defined based on the Euclidean distance. Given the two sets of neighbors, scDEED calculates cell i 's Euclidean distances to the ordered neighbors (from the closest to the farthest) in each set in the 2D embedding space, obtaining two distance vectors of length $x\% \times n$ (rounded to the closest integer). Finally, scDEED defines the reliability score as the Pearson correlation of the two distance vectors. That is, each cell's reliability score ranges from -1 to 1; a higher reliability score indicates a better agreement between the cell's ordered neighbors before and after the 2D embedding. We use the Pearson correlation because the actual values of the Euclidean distances in the 2D embedding space matter (for our visualization and interpretation), not just the ranks of the Euclidean distances used in the Spearman correlation.

Figure S15: a, PCA plots colored by cell type of the inDrops dataset (left), a permuted version of the inDrops dataset (middle), and a second permuted version of the inDrops dataset (right). b, t-SNE plots at the original perplexity of 40 in the same order as a. c, gene expression heatmaps with the cells grouped by cell type. Note that we truncated scaled expression values above 1 (2% of values) to make a more informative heatmap.

2. The authors only used gene expression heatmaps to demonstrate that scDEED could generate more reliable 2D embedding results, which are not convincing. In addition, the heatmap results shown by the authors are also questionable. For example, both the dubious cell and trustworthy cells shown in Supplementary Figure 1a were scattered in different cell clusters, how could the dubious cells or trustworthy cells from different cell clusters have highly similar gene expression profile in Supplementary Figure 1b?

Response:

First, we respectfully disagree with the reviewer regarding the similarity in gene expression. We would like to clarify the possible misunderstanding. In fact, we didn't state that dubious cells (or trustworthy cells) from different cell clusters have similar gene expression profiles; instead, we argued that even in one cluster, the cell embeddings that are labeled as dubious and trustworthy have different gene expression profiles. This was what we demonstrated based on the heatmap in Supplementary Figure S1b, the trustworthy cell embeddings (blue) and dubious cell embeddings (red) in the *nematocyte* cluster have distinct expression profiles.

Second, we agree with the reviewer that the highlighted dubious and trustworthy cells in the *nematocyte* cluster are scattered. We believe this result highlights the functionality of scDEED for showing that the original t-SNE perplexity of 40 is not optimized. The reason is that the highly similar gene expression within the trustworthy or dubious *nematocyte* embeddings suggest two cohesive clusters (Supplementary Figure S1b), but this was not reflected by the original t-SNE embedding (Supplementary Figure S1a).

Supplementary Figure S1 is pasted below for the reviewer's reference.

Figure S1. a, Comparative t-SNE plots at the original perplexity 40 with dubious (left) and trustworthy cell embeddings (right) in the *nematocytes*. **b,** Gene expression heatmap of the highlighted cells in **a**.

3. The authors only chose cells with > 95 th percentiles of null reliability scores as trustworthy embeddings, and treat cells with < 5 th percentile of the null reliability scores as dubious embeddings, then how do they deal with the vast majority of cells with null reliability scores between 5th ~ 95th percentiles? Why in most figures, the proportion of trustworthy embeddings is much larger than 95th percentiles?

Response:

We thank the reviewer for this comment, but we are afraid that the reviewer had a misunderstanding of the percentiles. Here we would like to re-elaborate on how the percentiles are generated. The 5th and 95th thresholding percentiles are defined using the *null reliability scores*, which are the correlations calculated from the *permuted* dataset (Step 5 in the **Methods section “The scDEED algorithm”**). Therefore, these two percentiles have nothing to do with the cells’ reliability scores calculated from the original dataset. As a result, we do not expect 5% of the cells’ reliability scores to be greater than the 95th percentile (or less than the 5th percentile) of the null reliability scores, since we are looking at two different distributions (null reliability score distribution from the permuted dataset vs. cells’ reliability score distribution from the original dataset). Hence, it is not surprising that most of the cell embeddings can be found as trustworthy.

To clarify this point, we have recreated a more comprehensive **Figure 1** (pasted on the next page), which provides greater detail and better clarity in illustrating scDEED’s algorithm. We have also added a new **Results section “A brief description of the scDEED algorithm”** as a less technical description of the scDEED algorithm, with the relevant paragraph pasted below. We hope these changes will help avoid misconceptions.

To construct a null distribution of reliability scores, scDEED employs a permutation strategy: independently permuting every gene’s expression levels across cells. After the permutation, all cells’ relationships are disrupted; all cells become exchangeable, and each cells’ neighbors become a random set of all cells. As a result, all cell’s reliability scores computed on the permuted data form a null distribution, i.e., the distribution of a cell’s reliability score if 2D embedding disrupts the cell’s neighbors in the pre-embedding space and randomly assigns neighbors to the cell in the 2D-embedding space (Methods). Based on this null distribution, two reliability score cutoffs are defined: (1) a trustworthy cutoff corresponding to the 95th percentile of the null distribution and (2) a dubious cutoff corresponding to the 5th percentile of the null distribution. Hence, for each cell in the original data, scDEED labels its 2D embedding as trustworthy if its reliability score is greater than or equal to the trustworthy cutoff. Conversely, if the cell’s reliability score is less than or equal to the dubious cutoff, its 2D embedding is labeled as dubious. Cells that do not meet these criteria remain unlabelled. It is worth noting that the percentage of cell embeddings labeled as trustworthy (or dubious) is not necessarily 5%, as the percentile cutoffs are defined based on the null distribution rather than the reliability scores computed from real data.

Figure 1. Illustration of the two functionalities of scDEED. Functionality I decides whether each cell has a *trustworthy* or *dubious* embedding by calculating a *reliability score*, which is defined as the Pearson correlation between the cell's distances to its closest 50% neighboring cells in the 2D-embedding space and the same cell's distances to its closest 50% neighboring cells in the pre-embedding space (with the distances in each space ordered from the 1st neighbor to the $[n/2]$ th neighbor, where n is the total number of cells). Compared with a *null distribution* of reliability scores, obtained through permutation, cell 1's reliability score (marked by the purple star) falls into the highest 5%, so it has a trustworthy embedding; in contrast, cell 2's reliability score (marked by the orange star) falls into the lowest 5%, so it has a dubious embedding. Enabled by functionality I, functionality II optimizes the hyperparameter setting of an embedding method (e.g. t-SNE and UMAP) by minimizing the number of dubious embeddings.

To further clarify, we consider the cell embeddings with reliability scores between the 5th and 95th percentiles of null reliability scores to be intermediate; these embeddings are not extreme enough to be considered either as dubious or trustworthy. We therefore do not use these cell embeddings for the optimization of hyperparameters. It is worth noting that we leave the thresholding percentiles (set as 5% and 95% by default) as optional arguments in scDEED, so depending on each user's level of conservativeness, they can adjust the definitions of dubious and trustworthy embeddings.

4. For the same dataset shown in Figure 2c and Supplementary Figure 1a, why did the number of trustworthy cells differ so greatly with the same parameter setting?

Response:

In fact, Supplementary Figure 1a only highlights the trustworthy and dubious cell embeddings in the *nematocyte* cluster, while Figure 2c shows the trustworthy cell embeddings in the entire dataset, across different cell types. Both figures are pasted below for the reviewer's reference.

Figure S1. **a**, Comparative t-SNE plots at the original perplexity 40 with dubious (left) and trustworthy cell embeddings (right) in the *nematocytes*. **b**, Gene expression heatmap of the highlighted cells in **a**.

Figure 2. Original t-SNE embeddings and t-SNE embeddings optimized by scDEED on the *Hydra* dataset. **a**, t-SNE plot of the *Hydra* dataset at the perplexity 40 used in the original study. **b–c**, Dubious cell embeddings (**b**) and trustworthy cell embeddings (**c**) defined by scDEED at the perplexity 40. **d**, t-SNE plot of the *Hydra* dataset at the perplexity 230 optimized by scDEED. **e–f**, Dubious cell embeddings (**e**) and trustworthy cell embeddings (**f**) defined by scDEED at the perplexity 230.

Reviewer #2 (Remarks to the Author):

t-SNE and UMAP are arguably the most widely used data visualization methods in single-cell genomics data analysis. While they are effective in manifesting clustered patterns of high-dimensional single-cell genomics data, such embedding methods are notoriously confusing for biomedical researchers to interpret due to the non-linear nature of the t-SNE and UMAP dimensionality reduction algorithms, resulting in unpreserved distances between data points after embedding. A computational approach to quantify the level of reliability or trustworthiness of such embedding methods is crucial and needed in the field.

In this manuscript, Xia et al. present a statistical method for identifying dubious embeddings for single cells in 2D visualizations such as t-SNE and UMAP. This method calculates a correlation score for each data point in an embedding dataset to quantify the embedding's reliability and identifies dubious data points with untrustworthy embeddings based on a statistical model with permutation-generated null distributions. This method can be used to optimize hyperparameter settings in embedding methods. The method is well designed with simple and clear ideas as well as rigorous statistical assessments. The authors have clearly shown the effectiveness of this method in several real scRNA-seq datasets in detecting unreliably placed cells or improving embedding analysis. The manuscript is clearly written, and the data presented are convincing. Based on the data presented in the manuscript, I believe every researcher who analyzes single-cell genomics data should use scDEED to evaluate and optimize t-SNE or UMAP embedding for reliable results before presenting the 2D visualization data. I have no doubt that this timely method should and will be widely used in the field.

We thank the reviewer for their enthusiastic comments. We also hope that scDEED will be a widely used tool in the single-cell field.

I have some minor comments that the authors may want to consider for improving the clarity of the manuscript.

1. For Figure 1, it might be beneficial to include additional graphs to provide a more detailed representation of the scDEED procedure. For example, scatter plots showing the original and embedding distances (in x and y) of a cell to its $n/2$ neighbors for Pearson correlation calculation could be added for cell 1 and cell 2 to demonstrate the different levels of correlation between the two example cells. Additionally, plotting the reliability score distributions for both real data and permutation data would also be helpful to show the procedure.

2. In Figure 1, the legend currently showing "1st neighbor" and "[$n/2$]th neighbor" is potentially confusing and misleading. Instead, it would be clearer to show a full color palette indicating the continuous color scale from the 1st neighbor to the $n/2$ th neighbor. As there are more than 2 colors involved in representing the full spectrum of the $n/2$ neighbors, a color palette would be more appropriate.

Response:

We thank the reviewer for their constructive suggestions regarding Figure 1. We fully agree and have redesigned Figure 1 based on the reviewer's suggestions for better clarity. Specifically, the updated Figure 1 includes the following changes: 1) two colored vectors were added to represent the two vectors of 2D Euclidean distances to a target cell: one vector based on the ordered neighbors in the 2D embedding space and the other vector based on the ordered neighbors in the pre-embedding space. The more similar the two vectors, the higher the Pearson correlation between them, and thus, the higher the reliability score of the target cell; 2) a row panel for permuted data was added to illustrate the calculation of the null reliability scores; 3) a distribution of the null reliability scores was plotted with marks for the reliability scores of cell 1 and cell 2, providing readers with more insights into the definition of dubious and trustworthy embeddings; 4) a color key with a color gradient was added to represent the neighbor order from the 1st neighbor to the $\lfloor n/2 \rfloor$ th neighbor.

For the reviewer's convenience, we have copied the **new Figure 1** on the next page.

Figure 1. Illustration of the two functionalities of scDEED. Functionality I decides whether each cell has a *trustworthy* or *dubious* embedding by calculating a *reliability score*, which is defined as the Pearson correlation between the cell's distances to its closest 50% neighboring cells in the 2D-embedding space and the same cell's distances to its closest 50% neighboring cells in the pre-embedding space (with the distances in each space ordered from the 1st neighbor to the $[n/2]$ th neighbor, where n is the total number of cells). Compared with a *null distribution* of reliability scores, obtained through permutation, cell 1's reliability score (marked by the purple star) falls into the highest 5%, so it has a trustworthy embedding; in contrast, cell 2's reliability score (marked by the orange star) falls into the lowest 5%, so it has a dubious embedding. Enabled by functionality I, functionality II optimizes the hyperparameter setting of an embedding method (e.g. t-SNE and UMAP) by minimizing the number of dubious embeddings.

3. *In Figures 2, 6, & 8, Panels a & d in each figure, the cell type labels on the t-SNE or UMAP plots are difficult to read. It is recommended that the cell types be labeled on a separate color legend.*

Response:

We thank the reviewer for the suggestion. We have added accordingly three new figures to the **Supplementary Appendix**, where cell types are labeled on a separate color legend. We have copied the **newly added Supplementary Figures A1–A3** on the next few pages.

Figure A1: Enlarged version of Figure 2A and 2D (the *Hydra* dataset) with a full legend of cell types.

Figure A2: Enlarged version of Figure 6A and 6D (the Alveolar dataset) with a full legend of cell types.

Figure A3: Enlarged version of Figure 8A and 8D (the *Samusik* dataset) with a full legend of cell types.

4. In Figures 3c, 3f, 3i, 5c, 5f, 5i, 7c, and 7f, the color scales of the heatmaps can be different for different datasets (while being the same for the same dataset/figure), to increase the contrast of the heatmaps and improve visualization.

Response:

We thank the reviewer for this constructive suggestion. We have changed the color scales of the heatmaps to increase the contrast and improve the visualization. We copied the **updated figures** on the next few pages.

Figure 3. Evaluation of t-SNE embeddings optimized by scDEED on the *Hydra* dataset. a–b, Comparative t-SNE plots with the *ecEP_sc*, *trustworthy cell embeddings in neuron ec1*, and *dubious cell embeddings in neuron ec1* highlighted, at the original perplexity 40 (a) and the perplexity 230 optimized by scDEED (b). **c,** Gene expression heatmap of the highlighted cells in a and b, where the cells are ordered by the default hierarchical clustering found by the R function `heatmap.2()`. **d–e,** Comparative t-SNE plots with the *neuron ec1*, *neuron ec3*, and *ecEP_sc* cells highlighted. At the original perplexity 40 (d), the *neuron ec1* cells are in two separate clusters and have similarly short distances as the *neuron ec3* cells have to the *ecEP_sc* cells; at the optimized perplexity 230 (e), the *neuron ec1* and *neuron ec3* cells are unified as one cluster far away from the *ecEP_sc* cells. **f,** Gene expression heatmaps of the highlighted cells in e and f, where the cells are ordered by the default hierarchical clustering found by the R function `heatmap.2()`. **g–h,** Comparative t-SNE plots with the *neuron ec1*, *neuron ec2*, *neuron ec3*, *neuron ec4*, *neuron ec5*, and *battery cell 2 (mp)* cells highlighted. At the original perplexity 40 (g), the *neuron ec1*, *neuron ec2*, *neuron ec3*, *neuron ec4*, and *neuron ec5* cells are in distinct clusters surrounding the *battery cell 2 (mp)* cells; at the optimized perplexity 210 (h), the five *neuron ec* clusters are unified as one cluster far away from the *battery cell 2 (mp)* cells. **i,** Gene expression heatmaps of the highlighted cells in g and h, where the cells are ordered by the default hierarchical clustering found by the R function `heatmap.2()`.

Figure 5: Evaluation of cluster locations at the original perplexity and the perplexity optimized by scDEED on the CAR-T dataset. **a–b**, Comparative t-SNE plots of *cluster 7*'s dubious and trustworthy cell embeddings at the original perplexity 30 (**a**) and the perplexity 750 optimized by scDEED (**b**). **c**, Gene expression heatmap of highlighted cells in **a** and **b**. **d–e**, Comparative t-SNE plots of *clusters 6, 7, and 14* at the perplexity 30 in the original study (**d**) and the perplexity 750 optimized by scDEED (**e**). We have recolored *cluster 7* for better visualization in **d–f**. **f**, Gene expression heatmap of the highlighted cells in **d** and **e**, where cells are ordered by the default hierarchical clustering found by the R function `heatmap()`. **g–h**, Comparative t-SNE plots of *clusters 3, 4, 9 and 11* at the perplexity 30 in the original study (**g**) and the perplexity 750 optimized by scDEED (**h**). We have recolored the clusters for better visualization in **g–i**. **i**, Gene expression heatmap of the highlighted cells in **g** and **h**.

Figure 7: Evaluation of cluster locations at the original hyperparameters and the hyperparameters jointly optimized by scDEED on the *Alveolar* dataset. **a–b**, Comparative UMAP plots of the *Alveolar* dataset with the *lec*, *b_cells*, *vcam1_veec*, and *vec* cells highlighted at the original hyperparameters of $\text{min.dist} = 0.3$ and $\text{n.neighbors} = 10$ (**a**) and the hyperparameters of $\text{min.dist} = 0.7$ and $\text{n.neighbors} = 6$ jointly optimized by scDEED (**b**). **c**, Gene expression heatmap of the highlighted cells in **a** and **b**. **d–e**, Comparative UMAP plots of the *Alveolar* dataset with the *lec*, *b_cells*, and *t_cells* cells highlighted at the original hyperparameters (**d**) and the hyperparameters jointly optimized by scDEED (**e**). **f**, Gene expression heatmap of the highlighted cells in **d** and **e**. Note we randomly downsampled *b_cells* and *t_cells* (from 911 and 2709 cells, respectively) so that each cluster has 256 cells (same as the number of cells in *lec*) to make a visually informative heatmap.

5. The authors performed a sensitivity analysis of scDEED to its only hyperparameter, x , the percentage of closest neighbors. However, the analysis only explored x values for 40, 50, and 60. It may be interesting for the authors to comment on the possibility of using much smaller or larger x . As the actual distances can be very large across cells from different cell types/clusters, and the Pearson correlation coefficient is sensitive to outliers, if a user only cares about preserving distances within close neighbors (e.g., cells in the same cluster/cell type), would using a very small x (e.g., 5, 10, 20) still be appropriate for calculating the reliability score? Any comments or recommendations from the authors would be helpful.

Response:

We thank the reviewer for this critical point. To address this point, we have added a new Methods section to discuss the choice of x , the similarity percent. Specifically, we have conducted a comprehensive sensitivity analysis of x , by extending the search grid to include $x = 5, 20, 35, 65, 80,$ and 95 . In summary, we observed that using too small or large values of x is generally ineffective. We copied the **new Methods section** and the **new Supplementary Figures** below.

Sensitivity analysis of scDEED's only hyperparameter (similarity percent x : the percentage of closest neighbors)

Steps 4 and 5 of the scDEED algorithm require the only hyperparameter of scDEED, x , the "similarity percent" (i.e., the percentage of closest neighbors, or the neighborhood size). The default value is $x = 50$, meaning that half of all cells are considered as neighbors. Intuitively, a smaller value of x defines a smaller neighborhood size and would thus place a greater emphasis on preserving local structures. To investigate the effect of x on the performance of scDEED, we examined $x = 5, 20, 35, 50, 65, 80,$ and 95 on the *Hydra* dataset with t-SNE as the embedding method.

First, we examined the numbers of dubious and trustworthy cell embeddings found at each x when applying scDEED under a range of t-SNE perplexity values (Supplementary Figure S16a–c). At a small similarity percent $x = 5$, the number of dubious cell embeddings was relatively stable across the perplexity values, an expected result as t-SNE was designed to preserve cells' local neighborhoods. At a large similarity percent $x = 95$, the number of trustworthy embeddings was smaller than at the other x values under most perplexity values; this result was also expected because t-SNE was not designed to preserve cells' global topology. Hence, using a too small or large x would not reflect t-SNE's ability to preserve mid-range neighbors or adjacent cell clusters' relative positions. This result justified the default value of $x = 50$ in scDEED.

Second, we observed that, at any x , as the perplexity value increased past a threshold (around 170 for the *Hydra* dataset), the number of dubious cells tended to stay stable and did not decrease further (Supplementary Figure S16a). This result provided evidence that scDEED

does not have a bias towards large perplexity values. Notably, at the original perplexity, too small or large x values found fewer dubious cell embeddings than the x values around the default $x = 50$ did, again implying that too small or large x values are unsuitable for detecting dubious cell embeddings (Supplementary Figure S16d). Most importantly, the x values around 50 resulted in similar optimized perplexity values, confirming the stability of scDEED with the default $x = 50$ (Supplementary Figure S16e).

To further explain the above observations, we examined the distributions of reliability scores in the permuted data and the original data at each x . We found the distributions of null reliability scores (i.e., reliability scores of permuted cells) to be stable across x values despite exhibiting a slight monotone shift to the right as x increased (Supplementary Figure S16f). On the original data, the distribution of reliability scores of unpermuted cells was most concentrated on lower scores when $x = 95$ (Supplementary Figure S16g). This explains why $x = 95$ found the fewest cell embeddings to be trustworthy.

Third, we examined the dubious or trustworthy cell embeddings detected by scDEED at $x = 5$, 50, or 95 from the original *Hydra* embeddings. Several clusters in the original embeddings were detected as dubious at both $x = 50$ and 95, but not at $x = 5$ (Supplementary Figure S17a). Also, only at $x = 5$, almost all cell embeddings were found as trustworthy (Supplementary Figure S17b). We believe that the different detection result at $x = 5$ was due to the fact that examining too small neighborhoods was ineffective in revealing the clusters with dubious positions relative to other clusters.

Fourth, we further examined the similarities of dubious cell embeddings detected at different x values given a perplexity value. We focused on dubious cell embeddings because their number is the criterion scDEED uses for optimization. We considered three perplexity values covering a wide range: perplexity 40 (used for the original embeddings), perplexity 230 (optimized by scDEED at $x = 50$), and perplexity 410 (the maximum candidate perplexity value in the scDEED package) (Supplementary Figure S18). Given each perplexity value, we used scDEED to detect a set of dubious cell embeddings at each x value; then we calculated the Jaccard index between the two sets for every pair of different x values (Supplementary Figure S18). We observed that the dubious embeddings detected at too small x values had little-to-no agreement with the dubious embeddings detected at other x values, an undesirable result as we would expect the dubious embeddings to be reasonably robust to the x value. In contrast, middle-to-high x values ($x = 50, 65,$ and 80) tended to have high agreement with each other, particularly $x = 50$. We also examined three x values close to 50 ($x = 40, 50, 60$) and confirmed that their respectively optimized visualizations were highly similar (Supplementary Figure S19).

In conclusion, the above sensitivity analysis results supported our default choice of $x = 50$. A similar rationale is described in [27], which found that effective dimension reduction required emphasis on mid-near neighbors.

Figure S16: a–c, Number of dubious (a), trustworthy (b), or intermediate (c) cell embeddings in the *Hydra* dataset found by scDEED with different “similarity percent” x values across t-SNE perplexity values. **d,** Number of dubious embeddings found using the original *Hydra* embeddings with different x values. **e,** Optimized perplexity found by scDEED with different x values. **f,** Distribution of the null reliability scores for different x found using the original perplexity 40 (left) and the respective optimized perplexity found with x (right). **g,** Distribution of the cells’ reliability scores for different x found using the original embeddings (left) and the embeddings produced by the respective optimized perplexity found with x (right).

Figure S17: a–b, t-SNE plots of the dubious (a) and trustworthy (b) cell embeddings found in the original embeddings of the *Hydra* dataset by scDEED using $x = 5$ (left), 50 (middle), and 95 (right).

Figure S18: Jaccard indices measuring the overlaps of dubious cell embeddings found at different x (similarity percent) values given a perplexity value of 40 (left; used for the original embeddings), 230 (middle; the scDEED optimized perplexity at $x = 50$) and 410 (right; the maximum candidate perplexity value in the scDEED package). Each horizontal axis value represents one x value, and each color/line represents another x value. Given a perplexity value, scDEED was applied to detect a set of dubious cell embeddings at each x value. A Jaccard index was calculated between every two sets of dubious cell embeddings, which corresponded to two different x values (one labeled by the horizontal axis and the other labeled by a color/line). Note that for each horizontal x value, we did not calculate the Jaccard index in the line corresponding to the same x value because the Jaccard index would be 1; for example, when the horizontal x value is 50, no Jaccard index is shown in the green line corresponding to $x = 50$.

Figure S19: Sensitivity analysis of scDEED. a–b, Comparative t-SNE plots with perplexities optimized by scDEED with the “similarity percent” hyperparameter (x , i.e., the percentage of closest neighbors, in Step 4 of the scDEED algorithm) set to $x = 40$, 50, and 60, corresponding to the three columns. Given each x , the perplexity was optimized by minimizing the number of dubious cell embeddings (the default approach in scDEED) (a) or the “kneedle” method (b). For easier visualization, the cell type labels are omitted, and the color key is the same as in Supplementary Appendix Figure A1.

Reviewer #3 (Remarks to the Author):

This manuscript introduces a permutation-based statistical method/procedure for assessing the reliability of scRNAseq data embedding by reporting the percentage of cells in the dataset with a reliable embedding. The reliability of each cell embedding is evaluated by how well the neighbor distances in the original data are preserved in the embedding compared with randomized cases. This work investigate an important problem in scRNAseq data interpretation and visualization. The current analysis still needs some more investigations to reveal some statistical properties of the permutation procedure. The evaluation and the presentation can also be improved.

Response:

We thank the reviewer for appreciating the importance of our work and for providing the constructive suggestions.

1. Lack of understanding of the potential inherent biases in the statistical measures.

1.1 Based on the results, scDEED tents to select larger tSNE smoothness parameter/Gaussian width/perplexity compared with commonly used choices (<50). Given large Gaussian widths often lead to over-smoothing/underfitting of the distance measures, it is very important to investigate the potential inherent biases in the application of the permutation. The plots in Figure S3 and S4 show that the number of dubious cell embeddings is much more in smaller perplexity (<100) than larger ones. Does this suggest a general trend that larger perplexity leads to smaller number of dubious cells and why?

Response:

We thank the reviewer for the observation and questions. In literature, some general rules of thumb, like the suggested perplexity equal to (the number of cells / 100) [1] or 1-10% of the total number of cells [2], have been proposed; they indicate that in general, the best perplexity should be larger than 40–50 whenever the dataset contains more than 4000–5000 cells, which is the case for most datasets we analyzed.

Nevertheless, it is not true that larger perplexity values always lead scDEED to detect a smaller number of dubious cells. In fact, with a grid of candidate perplexity values, the optimal perplexity selected by scDEED is not always the largest. To demonstrate this fact, we refer to **Supplementary Figures S3a, S4a, S16a, and S19** (pasted on the next few pages), which show the patterns between the number of dubious cell embeddings and the perplexity value. We observe that the trend is not monotone decreasing; instead, the number of dubious cell embeddings changes little past a certain perplexity value. For the *Hydra* and *CAR-T* datasets, the numbers of dubious cell embeddings vary little past the perplexity 170 (Supplementary Figure S3a and S16a for *Hydra*, Supplementary Figure S4a for *CAR-T*). In fact, the t-SNE visualization under the scDEED optimized perplexity (minimizing the number of dubious cell

embeddings) is highly similar to the t-SNE visualization under the perplexity selected by “kneedle” elbow point method (Supplementary Figure S3b, S4b, and S19). This validates the robustness of scDEED as the t-SNE visualization is largely unchanged when the number of dubious cell embeddings stays stable around the minimum. We have copied the relevant paragraph in the **newly added Methods section “Sensitivity analysis of scDEED’s only hyperparameter (similarity percent x: the percentage of closest neighbors)”** below.

Second, we observed that, at any x , as the perplexity value increased past a threshold (around 170 for the *Hydra* dataset), the number of dubious cells tended to stay stable and did not decrease further (Supplementary Figure S16a). This result provided evidence that scDEED does not have a bias towards large perplexity values. Notably, at the original perplexity, too small or large x values found fewer dubious cell embeddings than the x values around the default $x = 50$ did, again implying that too small or large x values are unsuitable for detecting dubious cell embeddings (Supplementary Figure S16d). Most importantly, the x values around 50 resulted in similar optimized perplexity values, confirming the stability of scDEED with the default $x = 50$ (Supplementary Figure S16e).

Figure S3. a, Plots of the number of dubious cell embeddings (the y-axis) versus perplexity (the x-axis) with the original and optimized perplexities highlighted. The optimized perplexities correspond to the minimum number of dubious embeddings (left), or the elbow point selected by the “kneedle” method [30] (right). b, Comparative t-SNE plots corresponding to the optimized perplexities 230 (left) and 170 (right).

Figure S4. **a**, Plots of the number of dubious cell embeddings (the y-axis) versus perplexity (the x-axis) with the original and optimized perplexities highlighted. The optimized perplexities correspond to the minimum number of dubious embeddings (left), or the elbow point selected by the “kneedle” method [30] (right). **b**, Comparative t-SNE plots corresponding to the optimized perplexities 750 (left) and 170 (right).

Figure S16: a–c, Number of dubious (a), trustworthy (b), or intermediate (c) cell embeddings in the *Hydra* dataset found by scDEED with different “similarity percent” x values across t-SNE perplexity values. **d**, Number of dubious embeddings found using the original *Hydra* embeddings with different x values. **e**, Optimized perplexity found by scDEED with different x values. **f**, Distribution of the null reliability scores for different x found using the original perplexity 40 (left) and the respective optimized perplexity found with x (right). **g**, Distribution of the cells’ reliability scores for different x found using the original embeddings (left) and the embeddings produced by the respective optimized perplexity found with x (right).

Figure S19: Sensitivity analysis of scDEED. a–b, Comparative t-SNE plots with perplexities optimized by scDEED with the “similarity percent” hyperparameter (x , i.e., the percentage of closest neighbors, in Step 4 of the scDEED algorithm) set to $x = 40, 50$, and 60 , corresponding to the three columns. Given each x , the perplexity was optimized by minimizing the number of dubious cell embeddings (the default approach in scDEED) (a) or the “kneedle” method (b). For easier visualization, the cell type labels are omitted, and the color key is the same as in Supplementary Appendix Figure A1.

1.2 One more potential bias to the definition of "dubious embedding" is the size of the cell clusters, e.g. in Figure 2b-2c, the dubious cells are almost all in the small clusters (such as "neuron en2"). It appears that the choice of $x\%$ of neighbors used for calculating the neighbor preservation is somewhat undecided. The plots in Figure S15 shows almost no difference with $x\% = 40\%$, 50% and 60% . What happens for smaller $x\%$ and larger $x\%$. Intuitively, $x\%$ should be related to the number and the size of the clusters?

Response:

We thank the reviewer for this critical point. To address this point, we have added a new Methods section to discuss the choice of x , the similarity percent. Specifically, we have conducted a comprehensive sensitivity analysis of x , by extending the search grid to include $x = 5, 20, 35, 65, 80$, and 95 . In summary, we observed that using too small or large values of x is generally ineffective. We copied the **new Methods section** and the **new Supplementary Figures** below.

Sensitivity analysis of scDEED's only hyperparameter (similarity percent x : the percentage of closest neighbors)

Steps 4 and 5 of the scDEED algorithm require the only hyperparameter of scDEED, x , the "similarity percent" (i.e., the percentage of closest neighbors, or the neighborhood size). The default value is $x = 50$, meaning that half of all cells are considered as neighbors. Intuitively, a smaller value of x defines a smaller neighborhood size and would thus place a greater emphasis on preserving local structures. To investigate the effect of x on the performance of scDEED, we examined $x = 5, 20, 35, 50, 65, 80$, and 95 on the *Hydra* dataset with t-SNE as the embedding method.

First, we examined the numbers of dubious and trustworthy cell embeddings found at each x when applying scDEED under a range of t-SNE perplexity values (Supplementary Figure S16a–c). At a small similarity percent $x = 5$, the number of dubious cell embeddings was relatively stable across the perplexity values, an expected result as t-SNE was designed to preserve cells' local neighborhoods. At a large similarity percent $x = 95$, the number of trustworthy embeddings was smaller than at the other x values under most perplexity values; this result was also expected because t-SNE was not designed to preserve cells' global topology. Hence, using a too small or large x would not reflect t-SNE's ability to preserve mid-range neighbors or adjacent cell clusters' relative positions. This result justified the default value of $x = 50$ in scDEED.

Second, we observed that, at any x , as the perplexity value increased past a threshold (around 170 for the *Hydra* dataset), the number of dubious cells tended to stay stable and did not decrease further (Supplementary Figure S16a). This result provided evidence that scDEED does not have a bias towards large perplexity values. Notably, at the original perplexity, too small or large x values found fewer dubious cell embeddings than the x values around the

default $x = 50$ did, again implying that too small or large x values are unsuitable for detecting dubious cell embeddings (Supplementary Figure S16d). Most importantly, the x values around 50 resulted in similar optimized perplexity values, confirming the stability of scDEED with the default $x = 50$ (Supplementary Figure S16e).

To further explain the above observations, we examined the distributions of reliability scores in the permuted data and the real data at each x . We found the distributions of null reliability scores (i.e., reliability scores of permuted cells) to be stable across x values despite exhibiting a slight monotone shift to the right as x increased (Supplementary Figure S16f). On the real data, the distribution of reliability scores of unpermuted cells was most concentrated on lower scores when $x = 95$ (Supplementary Figure S16g). This explains why $x = 95$ found the fewest cell embeddings to be trustworthy.

Third, we examined the dubious or trustworthy cell embeddings detected by scDEED at $x = 5$, 50, or 95 from the original *Hydra* embeddings. Several clusters in the original embeddings were detected as dubious at both $x = 50$ and 95, but not at $x = 5$ (Supplementary Figure S17a). Also, only at $x = 5$, almost all cell embeddings were found as trustworthy (Supplementary Figure S17b). We believe that the different detection result at $x = 5$ was due to the fact that examining too small neighborhoods was ineffective in revealing the clusters with dubious positions relative to other clusters.

Fourth, we further examined the similarities of dubious cell embeddings detected at different x values given a perplexity value. We focused on dubious cell embeddings because their number is the criterion scDEED uses for optimization. We considered three perplexity values covering a wide range: perplexity 40 (used for the original embeddings), perplexity 230 (optimized by scDEED at $x = 50$), and perplexity 410 (the maximum candidate perplexity value in the scDEED package) (Supplementary Figure S18). Given each perplexity value, we used scDEED to detect a set of dubious cell embeddings at each x value; then we calculated the Jaccard index between the two sets for every pair of x values (Supplementary Figure S18). We observed that the dubious embeddings detected at too small x values had little-to-no agreement with the dubious embeddings detected at other x values, an undesirable result as we would expect the dubious embeddings to be reasonably robust to the x value. In contrast, middle-to-high x values ($x = 50, 65, \text{ and } 80$) tended to have high agreement with each other, particularly $x = 50$. We also examined three x values close to 50 ($x = 40, 50, 60$) and confirmed that their respectively optimized visualizations were highly similar (Supplementary Figure S19).

In conclusion, the above sensitivity analysis results supported our default choice of $x = 50$. A similar rationale is described in [27], which found that effective dimension reduction required emphasis on mid-near neighbors.

Figure S16: a–c, Number of dubious (a), trustworthy (b), or intermediate (c) cell embeddings in the *Hydra* dataset found by scDEED with different “similarity percent” x values across t-SNE perplexity values. **d,** Number of dubious embeddings found using the original *Hydra* embeddings with different x values. **e,** Optimized perplexity found by scDEED with different x values. **f,** Distribution of the null reliability scores for different x found using the original perplexity 40 (left) and the respective optimized perplexity found with x (right). **g,** Distribution of the cells’ reliability scores for different x found using the original embeddings (left) and the embeddings produced by the respective optimized perplexity found with x (right).

Figure S17: a–b, t-SNE plots of the dubious (a) and trustworthy (b) cell embeddings found in the original embeddings of the *Hydra* dataset by scDEED using $x = 5$ (left), 50 (middle), and 95 (right).

Figure S18: Jaccard indices measuring the overlaps of dubious cell embeddings found at different x (similarity percent) values given a perplexity value of 40 (left; used for the original embeddings), 230 (middle; the scDEED optimized perplexity at $x = 50$) and 410 (right; the maximum candidate perplexity value in the scDEED package). Each horizontal axis value represents one x value, and each color/line represents another x value. Given a perplexity value, scDEED was applied to detect a set of dubious cell embeddings at each x value. A Jaccard index was calculated between every two sets of dubious cell embeddings, which corresponded to two different x values (one labeled by the horizontal axis and the other labeled by a color/line). Note that for each horizontal x value, we did not calculate the Jaccard index in the line corresponding to the same x value because the Jaccard index would be 1; for example, when the horizontal x value is 50, no Jaccard index is shown in the green line corresponding to $x = 50$.

Figure S19: Sensitivity analysis of scDEED. **a–b**, Comparative t-SNE plots with perplexities optimized by scDEED with the “similarity percent” hyperparameter (x , i.e., the percentage of closest neighbors, in Step 4 of the scDEED algorithm) set to $x = 40, 50$, and 60 , corresponding to the three columns. Given each x , the perplexity was optimized by minimizing the number of dubious cell embeddings (the default approach in scDEED) (**a**) or the “kneedle” method (**b**). For easier visualization, the cell type labels are omitted, and the color key is the same as in Supplementary Appendix Figure A1.

1.3 Another potential bias is the density of the gene expressions in different cell types. Would the density also affect the significance in the permutation? This can be problematic since scDEED might tend to pick cells with lower/higher density as more reliable?

Response:

We thank the reviewer for this question. Accordingly, we have examined the number of non-zero genes for cells in the *Hydra* dataset that were detected as dubious or trustworthy at varying perplexities. As illustrated in the following figure, we do not observe a tendency for scDEED to select cells with fewer genes expressed as dubious or trustworthy.

1.4 It will also be interesting to look into how the selection of the perplexity parameter changes the *t*-distributions of tSNE embedding. Does the parameter tuning leads to a better fitting *t*-distribution?

Response:

We thank the reviewer for this question. If our understanding of the question is correct (by fitting a *t* distribution, the reviewer meant the KL divergence used in t-SNE between the *t* distributions in the 2D embedding space and the Gaussian distributions in the pre-embedding space), then based on the literature, a larger perplexity value would decrease the KL divergence [3]. Hence, the KL divergence is unsuitable as a criterion for evaluating the perplexity value because it would favor larger perplexity values.

2. More evaluation will be helpful.

2.1. Current evaluation are mainly based interpreting the clustering results in each dataset. The KNN and KNC measures were only measured in the simulation data. Based on the ground-truth/human curated labels on the real datasets, does scDEED performs differently from EMBEDR?

Response:

We thank the reviewer for this suggestion. Accordingly, we have calculated the KNN and KNC on the *Hydra* dataset to quantitatively compare the t-SNE visualizations with the perplexities optimized by scDEED and EMBEDR respectively. As the two t-SNE visualizations are highly similar, the quantitative results are consistent with the visualizations. We have added the following **paragraph** to the Results and pasted **Supplementary Figure S12** on the next page for the reviewer's reference.

Although scDEED and EMBEDR found different optimized perplexity values on the *Hydra* dataset, the resulting t-SNE visualizations were highly similar (Supplementary Figure S12). Hence, we further compared the two t-SNE visualizations by evaluating two metrics regarding the preservation of neighboring information as in [20,44]. The first metric is the K-nearest neighbors (KNN), which reflects the preservation of local information, i.e., the average proportion of the 10 nearest neighbors in the pre-embedding space that remain in the set of 10 nearest neighbors in the 2D-embedding space (a proportion is calculated for every cell, and the average is taken over all cells' proportions). The second metric is the K-nearest clusters (KNC), which reflects the preservation of global information, i.e., the average proportion of the 4 nearest clusters in the pre-embedding space that remain in the set of 4 nearest clusters in the 2D-embedding space (a proportion is calculated for every cell, and the average is taken over all cells' proportions). For the *Hydra* dataset, the embeddings optimized by EMBEDR and the embeddings optimized by scDEED had the same KNC of 0.44, while the KNN was slightly better for scDEED (0.77) than EMBEDR (0.75). Thus, the two sets of optimized embeddings shared similar levels of information preservation.

Despite the similarity of the optimized embeddings produced by scDEED and EMBEDR on the *Hydra* dataset, scDEED far outperformed EMBEDR in terms of computational efficiency. Running without parallelization, scDEED completed the analysis (dubious embedding detection and perplexity optimization) in 4 hours, while EMBEDR finished in 18.5 hours using all available processors (the default setting in EMBEDR).

Figure S12. t-SNE visualizations of the *Hydra* dataset using the optimized perplexity of 210 found by scDEED (left) and the optimized perplexity of 410 found by EMBEDR (right).

2.2 Does the improved 2D embedding in visualization also quantitatively improve the clustering measured by ARI? It will be interesting check it as well.

Response:

For demonstration purposes, we used the *Human PBMC* dataset and calculated the ARI to compare the cell clusters found by spectral clustering on the 2D embeddings with the true cell type labels. We find that in general, the optimized embeddings lead to a higher ARI than the original embeddings. We have added the following **paragraph** to the Results and pasted the updated **Figure 10** with ARI values on the next page for the reviewer's reference.

Finally, we verified that the 2D cell embeddings became better aligned with the cell types after scDEED's optimization. Towards this goal, we applied spectral clustering to the original embeddings and scDEED's optimized embeddings to identify cell clusters and checked the agreement with the cell types by calculating the adjusted Rand index (ARI) [39]. We used spectral clustering because it is capable of identifying clusters of non-spherical shapes. As a measure of the agreement between cell cluster labels and cell type labels, ARI adjusts for the agreement due to random chance. We found that in five out of six cases (three scRNA-seq technologies with t-SNE and UMAP embeddings), the optimized embeddings led to higher ARIs than the original embeddings (Figure 10), suggesting that the optimized embeddings better represented the cell types.

Figure 10. Original t-SNE and UMAP embeddings and embeddings optimized by scDEED on the *Human PBMC* dataset. **a**, t-SNE and UMAP plots for the DropSeq dataset at the original hyperparameters, perplexity = 30 (left) and min.dist = 0.3 and n.neighbors = 30 (right). **b**, t-SNE and UMAP plots for the Dropseq dataset at the hyperparameters optimized by scDEED, perplexity = 290 (left) and min.dist = 0.5 and n.neighbors = 5 (right). **c**, t-SNE and UMAP plots for the inDrops dataset at the original hyperparameters, perplexity = 30 (left) and min.dist = 0.3 and n.neighbors = 30 (right). **d**, t-SNE and UMAP plots for the inDrops dataset at the hyperparameters optimized by scDEED, perplexity = 320 (left) and min.dist = 0.5 and n.neighbors = 80 (right). **e**, t-SNE and UMAP plots for the SeqWell dataset at the original hyperparameters, perplexity = 30 (left) and min.dist = 0.3 and n.neighbors = 30 (right). **f**, t-SNE and UMAP plots for the SeqWell dataset at the hyperparameters optimized by scDEED, perplexity = 140 (left) and min.dist = 0.2 and n.neighbors = 7 (right). Applying spectral clustering to identify cell clusters of the same number as the cell types in each set of embeddings, we found that in five out of six cases (three scRNA-seq technologies with t-SNE and UMAP embeddings), the optimized embeddings led to higher ARIs than the original embeddings, suggesting that the optimized embeddings better represented the cell types.

3. *The presentation needs some improvement.*

3.1 *Some important results for understanding the methods are discussed in the method section instead of the result section. It will be helpful to add a new section on the results of the statistical evaluation of the method.*

Response:

We thank the reviewer for this suggestion. Accordingly, we have added the following **new section** in the Results to introduce the scDEED algorithm.

A brief description of the scDEED algorithm

Figure 11 illustrates the scDEED method, which evaluates a 2D-embedding method (such as t-SNE and UMAP) by comparing each cell's neighbors in the pre-embedding space and the 2D-embedding space, with both sets of neighbors defined by the Euclidean distance and a pre-specified neighborhood size, i.e., the number of cells multiplied by the "similarity percent" (scDEED's only hyperparameter, set to 50% by default; see Methods for an investigation of the similarity percent value). In practical uses of t-SNE and UMAP (such as in the Seurat pipeline), the pre-embedding space is defined by the top 20–50 principal components (PCs) of log-transformed normalized gene expression levels. As each cell's Euclidean distances to 2D-embedding neighbors are crucial for visual inspection of the data, scDEED defines for each cell a "reliability score" to measure the consistency of the cell's 2D Euclidean distances to ordered neighbors before and after the embedding. Specifically, two ordered distance vectors are constructed for each cell: a pre-embedding distance vector and a 2D-embedding distance vector, whose lengths are both equal to the neighborhood size. The pre-embedding distance vector contains the 2D Euclidean distances between the cell and the pre-embedding neighbors, following the order of these neighbors from the closest to the farthest in the pre-embedding space. The 2D-embedding distance vector contains the 2D Euclidean distances between the cell and the 2D-embedding neighbors, following the order of these neighbors from the closest to the farthest in the 2D-embedding space. Then, the cell's reliability score is defined as the Pearson correlation between the two vectors.

To construct a null distribution of reliability scores, scDEED employs a permutation strategy: independently permuting every gene's expression levels across cells. After the permutation, all cells' relationships are disrupted; all cells become exchangeable, and each cells' neighbors become a random set of all cells. As a result, all cell's reliability scores computed on the permuted data form a null distribution, i.e., the distribution of a cell's reliability score if 2D embedding disrupts the cell's neighbors in the pre-embedding space and randomly assigns neighbors to the cell in the 2D-embedding space (Methods). Based on this null distribution, two reliability score cutoffs are defined: (1) a trustworthy cutoff corresponding to the 95th percentile of the null distribution and (2) a dubious cutoff corresponding to the 5th percentile of the null distribution. Hence, for each cell in the original data, scDEED labels its 2D embedding as trustworthy if its reliability score is greater than or equal to the trustworthy cutoff. Conversely, if

the cell's reliability score is less than or equal to the dubious cutoff, its 2D embedding is labeled as dubious. Cells that do not meet these criteria remain unlabelled. It is worth noting that the percentage of cell embeddings labeled as trustworthy (or dubious) is not necessarily 5%, as the percentile cutoffs are defined based on the null distribution rather than the reliability scores computed from real data.

scDEED's identification of cell embeddings as dubious or trustworthy allows users to identify potentially spurious cell clusters in the 2D-embedding space that are artifacts of the embedding process, rather than representing biologically meaningful cell types. It can also highlight clusters whose global positioning may be misleading. The lack of global preservation and the random positioning of clusters is a well-known issue in t-SNE and UMAP [20]. By identifying dubious cell embeddings, scDEED aims to address this issue. In the following sections, we will show examples of how the identification of dubious cell embeddings helps reveal dubious cell-type relationships in the 2D visualization.

To help users obtain a more trustworthy 2D visualization of data, scDEED provides an approach to optimize a 2D-embedding method's hyperparameters (e.g., perplexity for t-SNE; min.dist and n.neighbors for UMAP) via a grid search for the hyperparameter setting that minimizes the number of dubious cell embeddings.

We note that scDEED is a flexible algorithm applicable to any 2D-embedding method, not limited to t-SNE and UMAP. In our results, we focus on t-SNE and UMAP to showcase the effectiveness of scDEED in enhancing the reliability of 2D visualizations for drawing biologically meaningful conclusions.

3.2 The comparison with EMBEDR and the simulation should be presented first such that a reader can understand the result on each individual dataset better given this is a mainly a methodology paper?

Response:

We respectfully disagree with this suggestion for two reasons. First, we expect the predominant users of scDEED to be biologists who seek compelling evidence of its utility in real data analysis; otherwise, they will not be interested in the scDEED vs. EMBEDR comparison or the simulation results. Second, we need to present scDEED's results on the *Hydra* dataset, a major application case study of scDEED, before showing the scDEED vs. EMBEDR comparison because some of the comparison results used the *Hydra* dataset again.

References

1. Kobak D, Berens P. The art of using t-SNE for single-cell transcriptomics. *Nat Commun.* 2019;10: 5416.
2. Heiser CN, Lau KS. A Quantitative Framework for Evaluating Single-Cell Data Structure Preservation by Dimensionality Reduction Techniques. *Cell Reports.* 2020. p. 107576. doi:10.1016/j.celrep.2020.107576
3. Cao Y, Wang L. Automatic Selection of t-SNE Perplexity. *arXiv [cs.AI].* 2017. Available: <http://arxiv.org/abs/1708.03229>
4. Huang H, Wang Y, Rudin C, Browne EP. Towards a comprehensive evaluation of dimension reduction methods for transcriptomic data visualization. *Commun Biol.* 2022;5: 719.

Reviewer #1 (Remarks to the Author):

The authors have addressed my previous questions and the manuscript has been significantly improved. I now support the manuscript for publication.

Reviewer #2 (Remarks to the Author):

The authors have thoroughly addressed all my comments to the previous version of the manuscript. The revised manuscript has been much improved. I do not have any more concern, and recommend that the revised manuscript be accepted for publication.

Reviewer #3 (Remarks to the Author):

The additional supplementary figures are very helpful in demonstrating the sensitivity of the permutation to the choice of the perplexity for tSNE and optimality of hyperparameter x .

Below are some more minor clarifications needed,

* In question 1.4, the suggestion was to check if the optimal perplexity such as 170 is a proper choice for tSNE since the perplexity is related to the entropy of the conditional neighboring distributions computed with an assumption of t-distribution.

* For question 1.3, it seems around the good choices of perplexity, dubious cells seem to be sparser? It might be expected (or good) since these cells have less information for accurate embedding? A bit more discussion/clarification is needed.

Other than these, I have no more comments.

Responses to Reviewers' Comments on the Revision of "scDEED: a statistical method for detecting dubious 2D single-cell embeddings and optimizing t-SNE and UMAP hyperparameters"

We thank the three reviewers for dedicating their time to reviewing our initial paper and the revised version. Based on the comments from Reviewer 3, we have made additional changes to the Discussion section accordingly.

On the following pages, please find our point-by-point responses to the three reviewers' comments. The reviewers' comments are in *italics*. Newly added text in the revised manuscript is highlighted in blue.

Reviewer #1 (Remarks to the Author):

The authors have addressed my previous questions and the manuscript has been significantly improved. I now support the manuscript for publication.

Reviewer #2 (Remarks to the Author):

The authors have thoroughly addressed all my comments to the previous version of the manuscript. The revised manuscript has been much improved. I do not have any more concern, and recommend that the revised manuscript be accepted for publication.

Response:

We thank Reviewers #1 and #2 for their time and insightful comments that have helped us improve the figures and the Methods section of our manuscript.

Reviewer #3 (Remarks to the Author):

The additional supplementary figures are very helpful in demonstrating the sensitivity of the permutation to the choice of the perplexity for tSNE and optimality of hyperparameter x .

Below are some more minor clarifications needed,

** In question 1.4, the suggestion was to check if the optimal perplexity such as 170 is a proper choice for tSNE since the perplexity is related to the entropy of the conditional neighboring distributions computed with an assumption of t-distribution.*

Response:

We would like to clarify that the perplexity hyperparameter is related to the definition of the neighbor-picking probabilities (which the reviewer refers to as the “conditional neighboring distributions”) in the high-dimensional space (referred to as “high-dim probabilities”). However, in t-SNE, the t-distribution is not used to specify the high-dim probabilities. Rather, the t-distribution is used to specify the neighbor-picking probabilities in the low-dimensional space (referred to as “low-dim probabilities”). While different perplexity values lead to different high-dim probabilities, how well the low-dim probabilities approximate the high-dim probabilities is a separate question, whose answer depends on the optimization (which minimizes the KL divergence between the two sets of probabilities). This is the reason why in our previous response, we interpreted the reviewer’s previous comment about “fitting of the t-distribution” as the KL divergence, which decreases as the perplexity value increases [1]. If our previous interpretation is incorrect, then we do not understand how we could check if a perplexity value is a proper choice for t-SNE without more specific guidance from the reviewer. In the following paragraph, we will explain the t-SNE method details to justify our argument that the perplexity value is only related to the high-dim probabilities, not the t-distribution used for the low-dim probabilities.

To discuss this question, we use the notations defined in the original t-SNE paper [2]. As a brief review, assume we have n high-dimensional data points (cells in scDEED) represented by x_1, \dots, x_n . The goal of t-SNE is to obtain low-dimensional embeddings y_1, \dots, y_n (2D embeddings in scDEED) to approximate x_1, \dots, x_n . We define $p_{j|i}$ to be the probability that x_i picks x_j as its neighbor. Similarly, we define $q_{j|i}$ to be the probability that y_i picks y_j as its neighbor. Ideally, we would like to have $p_{j|i} = q_{j|i}$ for all $i, j = 1, \dots, n$ so that the neighboring information of data points is preserved in the low-dimensional embeddings. Hence, to find y_1, \dots, y_n , the goal is to minimize the KL divergence between the two n -value discrete probability distributions $P_i = (p_{1|i}, \dots, p_{n|i})^T$ and $Q_i = (q_{1|i}, \dots, q_{n|i})^T$ for each $i = 1, \dots, n$. The explicit definitions of $p_{j|i}$ and $q_{j|i}$ are

$$p_{j|i} = \frac{\exp\left(-\frac{\|x_i - x_j\|^2}{2\sigma_i^2}\right)}{\sum_{k \neq i} \exp\left(-\frac{\|x_i - x_k\|^2}{2\sigma_i^2}\right)} \quad (1)$$

$$q_{j|i} = \frac{\left(1 + \|y_i - y_j\|^2\right)^{-1}}{\sum_{k \neq i} \left(1 + \|y_i - y_k\|^2\right)^{-1}}. \quad (2)$$

The perplexity hyperparameter is required for specifying the σ_i parameter in (1). Specifically, the perplexity of P_i is defined as $Perp(P_i) = 2^{H(P_i)}$, where $H(P_i)$ is the Shannon entropy measured in bits. Hence, a larger perplexity value means a higher entropy of P_i (i.e., the n probabilities $p_{1|i}, \dots, p_{n|i}$ are more similar in value), so σ_i is larger. Hence, changing the perplexity value corresponds to changing $p_{j|i}$ for all $i, j = 1, \dots, n$.

Note that the t-distribution is only involved in (2), not in (1). Hence, the perplexity value does not affect the t-distribution density formula in (2). Instead, when the perplexity value changes, the high-dim probabilities $p_{1|i}, \dots, p_{n|i}$ change through (1), and the optimization (minimization of the KL divergence) finds a different set of y_1, \dots, y_n so that through (2) $q_{1|i}, \dots, q_{n|i}$ can cope with the changes in $p_{1|i}, \dots, p_{n|i}$. In summary, when the perplexity value changes, the low-dimensional embeddings y_1, \dots, y_n change accordingly, but this change is not through altering the t-distribution density formula.

We hope the above explanation clarifies why we do not understand the reviewer's comment that *"the perplexity is related to the entropy of the conditional neighboring distributions computed with an assumption of t-distribution."* The reason is that the perplexity is related to the entropy of the high-dim probabilities, which do not involve the t-distribution.

If the reviewer meant to use the low-dim probabilities $q_{1|i}, \dots, q_{n|i}$ to define some optimality of the perplexity, then we regard the optimality definition as an open problem that requires future research. We have added the following text to the last paragraph of the **Discussion** section.

"However, open questions remain. First, ... Second, it remains an open question to understand how the optimized hyperparameter setting is related to the algorithms of t-SNE and UMAP. For example, t-SNE uses the perplexity to define cells' neighbor-picking probabilities in the pre-embedding space, and t-SNE's optimization tries to find cells' 2D embeddings so that the cells' neighbor-picking probabilities are well preserved in the 2D-embedding space. Hence, an open question is whether the optimized perplexity leads to more reasonable neighbor-picking probabilities that better align with cells' topology in some sense."

* For question 1.3, it seems around the good choices of perplexity, dubious cells seem to be sparser? It might be expected (or good) since these cells have less information for accurate embedding? A bit more discussion/clarification is needed.

Response:

We thank the reviewer for this insightful question. We would like to clarify that the “dubious” or “trustworthy” classification applies only to the cell embeddings, not the cells themselves. In the *Hydra* dataset, around the optimal perplexity values (230-270, which have the same minimal $n = 128$ dubious cell embeddings, along with $n = 24,832$ trustworthy cell embeddings), the mean sparsity level of the cells with dubious embeddings is slightly lower, not higher than the mean sparsity level of the cells with trustworthy embeddings. However, the difference between the two mean sparsity levels is not statistically significant by the two-sided t-test (at perplexity 230, p-value = 0.208; at perplexity 250, p-value = 0.161; and at perplexity 270, p-value = 0.069). In contrast, at perplexities much lower than the optimal perplexity, the cells with dubious embeddings demonstrate a higher sparsity level than the cells with trustworthy embeddings. Based on this observation, we hypothesize that at the optimal perplexity, there are random differences in the sparsity level between the cells with dubious embeddings and those with trustworthy embeddings. That is, at the optimal perplexity, the dubious cell embeddings do not have much relevance to sparsity and are from a small random set of cells. Investigating this hypothesis is a future research question. We have added the following text to the last paragraph of the **Discussion** section.

“However, open questions remain. First, we lack an understanding of the difference between dubious embeddings and trustworthy embeddings in terms of gene expression profiles. Our empirical evidence suggests that the difference depends on the hyperparameter value(s) at which the dubious embeddings are detected. For example, in the *Hydra* dataset, at a low t-SNE perplexity, cells with dubious embeddings tend to exhibit higher sparsity compared to cells with trustworthy embeddings. However, this trend reverses after the perplexity surpasses the optimal perplexity selected by scDEED. At the optimal perplexity, cells with trustworthy or dubious embeddings demonstrate statistically similar levels of sparsity (as shown in Supplementary Figure 20). This result might indicate that the difference between dubious embeddings and trustworthy embeddings is more due to randomness instead of biological signals under the optimized hyperparameter(s). As a future direction, exploring how the difference changes along with the hyperparameter value(s) might offer an alternative way for choosing the hyperparameter value(s).”

Supplementary Fig. S5: The number of nonzero genes in the cells with dubious or trustworthy embeddings at each of the varying t-SNE perplexity values in the Hydra dataset. Each dot (and its accompanying half-length error bar) indicate the mean (and standard deviation) of the number of nonzero genes in the cells with dubious embeddings or trustworthy embeddings at a given perplexity level. .

Other than these, I have no more comments.

Response:

We would like to thank the reviewer for the time and consideration.

1. Cao Y, Wang L. Automatic Selection of t-SNE Perplexity. arXiv [cs.AI]. 2017. Available: <http://arxiv.org/abs/1708.03229>
2. Gmail L, Hinton G. Visualizing Data using t-SNE. 2008 [cited 21 Sep 2023]. Available: <https://www.jmlr.org/papers/volume9/vandermaaten08a/vandermaaten08a.pdf?fbclid>
3. Belkina AC, Ciccolella CO, Anno R, Halpert R, Spidlen J, Snyder-Cappione JE. Automated optimized parameters for T-distributed stochastic neighbor embedding improve visualization and analysis of large datasets. *Nat Commun.* 2019;10: 5415.
4. Kobak D, Linderman GC. Initialization is critical for preserving global data structure in both t-SNE and UMAP. *Nature biotechnology.* 2021. pp. 156–157.